# Traveling Waves Integrate Spatial Information Through Time

**Mozes Jacobs[1]    Roberto C. Budzinski[2]    Lyle Muller[2]    Demba Ba[1]    T. Anderson Keller[1]**

[1]The Kempner Institute for the Study of Natural and Artificial Intelligence, Harvard University

[2]Western University, Department of Mathematics, London, Ontario, Canada

mozesjacobs@g.harvard.edu, rbudzins@uwo.ca, lmuller2@uwo.ca,
demba@seas.harvard.edu, t.anderson.keller@gmail.com

## Abstract

**Traveling waves of neural activity are widely observed in the brain, but their precise computational function remains unclear. One prominent hypothesis is that they enable the transfer and integration of spatial information across neural populations. However, few computational models have explored how traveling waves might be harnessed to perform such integrative processing. Drawing inspiration from the famous "*Can one hear the shape of a drum?*" problem – which highlights how normal modes of wave dynamics encode geometric information – we investigate whether similar principles can be leveraged in artificial neural networks. Specifically, we introduce convolutional recurrent neural networks that learn to produce traveling waves in their hidden states in response to visual stimuli, enabling spatial integration. By then treating these wave-like activation sequences as visual representations themselves, we obtain a powerful representational space that outperforms local feed-forward networks on tasks requiring global spatial context. In particular, we observe that traveling waves effectively expand the receptive field of locally connected neurons, supporting long-range encoding and communication of information. We demonstrate that models equipped with this mechanism solve visual semantic segmentation tasks demanding global integration, significantly outperforming local feed-forward models and rivaling non-local U-Net models with fewer parameters. As a first step toward traveling-wave-based communication and visual representation in artificial networks, our findings suggest wave-dynamics may provide efficiency and training stability benefits, while simultaneously offering a new framework for connecting models to biological recordings of neural activity.**

**Keywords:** Traveling Waves; Oscillation; Information Integration

## Introduction

The propagation of traveling waves of neural activity has been measured on the surface of the brain from the earliest neural recordings (Adrian & Matthews, 1934; Goldman et al., 1949; Lilly, 1949; Mickle & Ades, 1953). Such waves have been measured to travel both locally and globally across cortical regions with a range of velocities (Reimer et al., 2010; Muller et al., 2016; Zhang et al., 2018). Stimulus-evoked traveling waves have been directly measured in visual cortex (Cowey, 1964) with increasingly sophisticated methodology from penetrating electrodes (Ebersole & Kaplan, 1981) to voltage-sensitive dye imaging (Muller et al., 2014), including in awake behaving

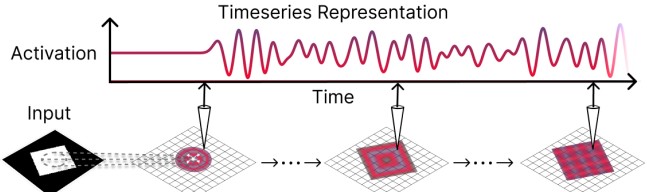

Figure 1: **Overview of traveling wave-based spatial information integration**. An input stimulus triggers an initial condition and sets the response properties of a lattice of neurons with both local input receptive fields and recurrent connectivity. This initial condition evolves over time under the recurrent wave dynamics, and the resulting timeseries at each neuron becomes a globally integrated representation of the visual stimuli.

primates (Davis et al., 2020). Driven by these observations, many theoretical arguments have been put forth to explain the functional roles of these dynamics. Examples include that they are relevant to predictive coding (Alamia & VanRullen, 2019), the representation of symmetries (Keller et al., 2024a), the consolidation of long-term memories (Muller et al., 2018), and the encoding of motion (Heitmann & Ermentrout, 2020).

Most relevant to this study, one often hypothesized role is that traveling waves serve as a mechanism for integration and transfer of information over long distances – a mechanism that is believed to play an important role specifically within visual cortex (Sato et al., 2012). For example, Kitano et al. (1994) demonstrated early on that local-field potential (LFP) responses of neurons in primary visual cortex could be elicited by stimuli far outside their classic retinotopic receptive fields with increased latency as a function of distance, implying a long-range distance-delayed integration of information. These findings were later reinforced by the intracellular subthreshold membrane potential recordings of Bringuier et al. (1999), denoting this extended receptive field the 'visually evoked synaptic integration field'. However, the hypothesized role of information transfer and integration extends beyond visual stimuli. For example, Rubino et al. (2007) found that beta frequency oscillations propagated spatially across the motor cortices of monkeys in preparation for movement, and that information about the visual target was directly encoded in these waves. Similarly, Besserve et al. (2015) used direction-specific causal information transfer metrics to demonstrate that traveling waves in the gamma frequency band are correlated with information transfer between different cortical regions; while Bhattacharya et al. (2022) showed that waves change direction during information

retrieval and processing in a working memory task.

Despite these promising observations however, it remains challenging to investigate these ideas computationally due to a lack of task-trainable artificial neural network models which exhibit traveling wave dynamics. In modern artificial neural networks, information is integrated and transmitted over spatial distances of an input (e.g. an image) or between 'tokens' of a sequence either via extremely deep convolutional neural networks (He et al., 2015), bottleneck/pooling layers (Ronneberger et al., 2015), or all-to-all connectivity as-in Transformers (Vaswani et al., 2023). Each of these approaches comes with its own computational complexity and expressivity limitations, and it is therefore of great interest to explore alternative methods to integrate disparate information in neural systems.

In this paper, we aim to make progress towards understanding the causal role of wave dynamics in the transfer of information by filling this modeling gap, and exploring the computational potential of wave-based models in task-relevant settings. To begin, we take inspiration from the famous mathematical question "*Can one hear the shape of a drum*", and explore if the techniques underlying this problem, namely the representation of global information through stationary solutions to wave-based dynamical systems, can be equivalently applied to extract global information from locally-connected recurrent neural network hidden states over time. In first part of this paper, we begin with a review of this problem and the associated formalism, outlining how we may construct trainable recurrent neural networks to leverage these ideas as a computational principle. On toy tasks, we demonstrate that these simple models do indeed match theoretical predictions (Figure 2), and that when slightly relaxed to allow for more flexible input encoding, they generate wave dynamics which enable the disentanglement of simple shapes in frequency space (Figures 3 & 4). In the second part of this paper, we use this intuition as motivation to build a suite of further relaxed, yet more computationally capable, convolutional recurrent neural networks (conv-RNNs), with inherently limited receptive field sizes in both their initial encoders and recurrent connections, and test them on more complex global-information processing (semantic segmentation) tasks. We demonstrate how by using the timeseries of each neuron's recurrent neural activity as our primary neural representation during training (schematized in Figure 1), such models are able to outperform other locally constrained models, and even rival the performance of some deeper convolutional models such as U-Nets, while often converging more consistently to favorable solutions – ultimately suggesting that traveling wave-based information integration may be an efficient and stable alternative to existing deep neural network spatial integration techniques.

## Motivation: Hearing the Shape of a Drum

To build intuition for how traveling waves may integrate information over space, we take inspiration from the famous mathematical question '*Can one hear the shape of a drum?*' posed by Mark Kac (1966). Simply put, this question asks whether the boundary conditions of an idealized drum head are uniquely

identified by the frequencies at which the drum head will vibrate. Intuitively, when one strikes a drum head, this initial disturbance will propagate outwards as a transient traveling wave until it reaches the fixed boundary conditions where it will reflect with a phase shift. This reflected wave will thus have *collected information* about the boundary, and serves to bring it back towards the center. These waves will eventually collide with other reflected waves from all edges of the shape, and combine in a superposition of wavefronts, eventually resulting in a solution which is a superposition of discrete normal modes which are constrained by the boundary of the shape itself.

At a high level, Kac's question investigates one specific mechanism by which traveling waves may be used to integrate global information, by evolving a spatially-local dynamical system (a wave equation) to a steady-state solution determined by global conditions (the fixed drum boundary). While there exist many other mechanisms by which traveling waves can be considered to transfer and combine information, such as through delay-line mechanisms (Jeffress, 1948) known to exist in the brainstems of owls (Carr & Konishi, 1988), or through more complex mechanisms such as interfering wave fronts (Gong & Van Leeuwen, 2009; Izhikevich & Hoppensteadt, 2009), the steady-state solution mechanism yields a powerful and well-understood starting point for us to begin building models.

Formally, a "drum" in this problem is considered to be a perfectly elastic two-dimensional membrane whose vertical displacement over space and time is denoted $u(x, y, t)$. The drum head is considered to be stretched under uniform tension to a boundary of shape $\Omega$, such that its dynamics satisfy the two-dimensional wave equation with constant wave-speed $c$:

$$\frac{\partial^2 u}{\partial t^2} = c^2 \nabla^2 u = c^2 \left( \frac{\partial^2 u}{\partial x^2} + \frac{\partial^2 u}{\partial y^2} \right). \tag{1}$$

In the original problem, the drum head is constrained to be 'clamped' to 0 displacement on the boundary, known as a Dirichlet boundary condition. This is often written as $u|_{\partial \Omega} = 0$. Since we are interested in steady-state oscillatory solutions, we can assert they must take the form of 'normal modes' $\phi_k(x, y)$ with associated oscillation frequencies $\omega_k$:

$$u(x, y, t) = \phi_k(x, y) \cos(\omega_k t). \tag{2}$$

Plugging this into Equation 1, we see the solutions must satisfy:

$$\nabla^2 \phi_k(x, y) = -\lambda_k^2 \phi_k(x, y), \quad \text{where} \quad \lambda_k^2 = \frac{\omega_k^2}{c^2}. \tag{3}$$

In this form, it is clear that $\lambda_k$ is an eigenvalue of the Laplacian operator acting on the surface $u$. Succinctly then, the question posed by Kac (1966), is if full set of eigenvalues $\{\lambda_1, \lambda_2, \ldots\}$ (called the eigenspectrum) of the Laplacian operating on a given boundary is sufficient to uniquely identify all two-dimensional boundaries. At the time of posing the question, it was known that the area of a drum-head could be deduced from its eigenspectrum uniquely; however, it took more than 25 years for researchers to find counter examples of drum

heads that could not be distinguished by their eigenspectra (Gordon et al., 1992), while later work from Zelditch (1999) was able to precisely characterize a class of shapes which are uniquely identifiable. Overall, these results demonstrated that the amount of unique geometric information in spectral representations is significant, and in most cases, aside from the pathological examples, most shapes are not 'isospectral'.

**The Solution for Square Drums**   For a square drum of side length $L$, the above boundary value problem has a well-known simple solution (Bérard & Helffer, 2014). Specifically, the normal modes and corresponding Laplacian eigenvalues are:

$$\phi_{m,n}(x,y) = \sin\left(\frac{m\pi}{L}x\right)\sin\left(\frac{n\pi}{L}y\right), \quad m,n = 1,2,3,\ldots \quad (4)$$

$$\lambda_{m,n} = \sqrt{\left(\frac{m\pi}{L}\right)^2 + \left(\frac{n\pi}{L}\right)^2}. \quad (5)$$

We can quickly verify that indeed, these modes are all zero at the boundary locations of the square since $\sin(\frac{n\pi}{L}x) = 0$ when $x = 0$, and $\sin(\frac{n\pi}{L}x) = 0$ when $x = L$. From Equation 3, the oscillation frequencies are $\omega_{m,n} = c\lambda_{m,n} = c\frac{\pi}{L}\sqrt{m^2+n^2}$. Therefore, the *lowest* resonant frequency of a square drum is $\omega_{1,1} = c\frac{\pi}{L}\sqrt{2}$, measured in radians per second, following our intuition that larger drums (larger $L$) produce lower pitches ($\omega$).

To validate our main idea that this mechanism for global information integration may be simulated reasonably in a recurrent neural network, in the following we implement a simple RNN model which emulates wave dynamics, and measure if the Fourier transform of the resulting hidden state dynamics inside the drum exhibits these fundamental frequencies.

**Emulation in a Recurrent Neural Network**   To simulate the above equation in an RNN, we observe that the wave equation (Equation 1) can be discretized over space and time to yield a set of equations which are very reminiscent of an RNN. This is the same approach taken by Keller et al. (2024b) for the first order one-way wave equation, and Rusch & Mishra (2021) for a network of coupled oscillators, but adapted to the standard 2D wave equation. Explicitly, we construct an RNN to accurately numerically integrate the wave equation using Verlet integration, yielding the following set of updates for the hidden state $\mathbf{h}$ and the associated coupled velocity state $\mathbf{v}$:

$$\mathbf{v}_{t+1/2} = \mathbf{v}_t + \frac{1}{2}\Delta t \cdot K_{\nabla^2} \star \mathbf{h}_t, \quad \mathbf{h}_{t+1} = \mathbf{h}_t + \Delta t \cdot \mathbf{v}_{t+1/2} \quad (6)$$

$$\mathbf{v}_{t+1} = \mathbf{v}_{t+1/2} + \frac{1}{2}\Delta t \cdot K_{\nabla^2} \star \mathbf{h}_{t+1} \quad (7)$$

Where $\mathbf{h} \in \mathbb{R}^{H \times W}$ is defined to have 2 spatial dimensions, $\star$ denotes convolution over these dimensions, and $K_{\nabla^2}$ is the five-point stencil for the discrete Laplacian operator in 2D:

$$K_{\nabla^2} = \begin{bmatrix} 0 & 1 & 0 \\ 1 & -4 & 1 \\ 0 & 1 & 0 \end{bmatrix}. \quad (8)$$

The most straightforward manner to then provide 'the drum as input' to the RNN, is to treat it as existing on a discretized

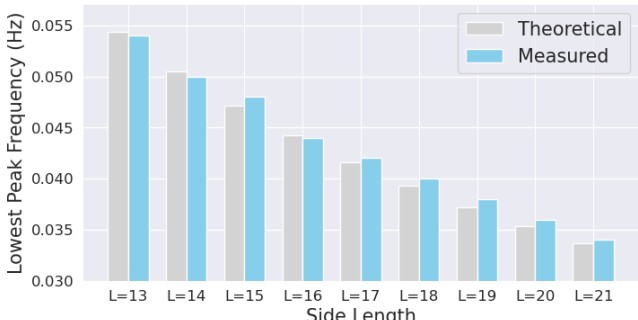

Figure 2: **Waves-RNNs generate theoretical frequencies.** Theoretical fundamental frequencies in Hz ($\frac{cycles}{sec}$) for square drum heads of different side lengths L, compared with the measured lowest peak frequencies of a wave-based RNN which uses the square input to determine it's recurrent dynamics.

grid (like an image), and map each spatial location (x,y) to a corresponding neuron $h_{x,y}$. We then emulate an idealized learned encoder by clamping the values of neurons at the boundary of the square (and outside) to zero. Explicitly, $h_{x,y} = 0 \ \forall \ \{x,y\} \in \Omega$. We can then provide an initial condition by setting the hidden state at the center of the drum ($h_{c_x,c_y}$) to a displacement of 1 with all other locations set to 0, and allow the dynamics above to unfold over time.

In Figure 2, we present the results of this experiment. We vary the square side length $L$ from 13 to 21, and for each $L$, we compute the theoretical fundamental frequency (in Hz) as $\omega_{1,1} = c\sqrt{2}/(2L)$. The corresponding lowest peak frequency is then measured from the Fourier transform of the hidden state dynamics at $h_{c_x,c_y}$ over 40,000 timesteps ($\Delta t = 0.025$) and plotted on the y-axis. The wave-based RNN's results align almost perfectly with theoretical predictions, with minor deviations likely due to numerical integration limitations.

## Experiments: Semantic Segmentation

In the following, we take the intuition gathered from the previous theoretical motivation and apply it to study if traveling waves can be used in locally constrained recurrent neural network architectures to solve a task requiring global information integration. In particular, on all experiments, the task is semantic segmentation, where each pixel of the original image must be classified as either background, or one of the classes from the dataset, and models are trained to minimize a pixel-wise cross-entropy loss. Crucially, all locally restricted models make use of shallow convolutional encoders (with $3 \times 3$ kernels), convolutional recurrent connections, and pixel-local decoders, ensuring that the spatial receptive field of each neuron in a single feed-forward pass is limited to be significancy less than the inherent length scale of features necessary to identify class labels in each dataset (visualized in Figure 7) – meaning that if the network solves the task, it must be integrating global information through recurrent connections. As baselines, we compare with CNN models of various depths, from 2 to 32 layers, which thereby have receptive fields which span from local to global with respect to the image. On the final more complex

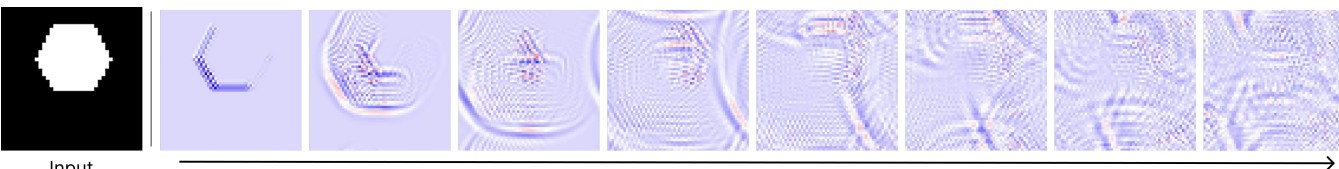

Input        RNN Hidden States Over Time

**Figure 3: Waves propagate differently inside and outside shapes, integrating global shape information to the interior.** Sequence of hidden states of an oscillator model (NWM) trained to classify pixels of polygon images based on the number of sides using only local encoders and recurrent connections. We see the model has learned to use differing natural frequencies inside and outside the shape to induce soft boundaries, causing reflection, thereby yielding different internal dynamics based on shape.

datasets, we additionally compare with a more advanced U-Net architecture (Ronneberger et al., 2015) which uses simultaneous depth and a spatial bottleneck to transmit information globally. We refer readers to the supplementary material for full training details. The full code and video visualizations for the results in this paper are available at: `https://github.com/KempnerInstitute/traveling-waves-integrate`

## Datasets

To validate the core idea that traveling waves can be used to transmit information over large spatial distances, and that this global information can be decoded in a task-relevant manner, we primarily employ four datasets:

**Polygons:** First, we consider a simple dataset composed of white polygons on black backgrounds, where the classes are given by the number of sides of the polygons. The examples are synthetic $75 \times 75$ pixel grayscale images with 1 to 2 polygons, each with 3 to 6 edges roughly circumscribed within circles with radii of 15 to 20 pixels. On this dataset, the angle of the corners of the shape are sufficient to correctly classify those patches, but this information must then be transferred to the center of the shape for correct segmentation of the interior.

**Tetrominoes:** As a second slightly more complex dataset that has been employed in prior segmentation work (Miyato et al., 2024), we employ own re-implementation of the Tetrominoes dataset (Kabra et al., 2019), where each image is composed of 1 to 5 'Tetris' like blocks of varying shapes and colors arranged on a black background. In detail, there are 6 distinct classes of objects from 14-28 pixels long. The increased complexity of shapes and number of objects per image increases the difficulty of this dataset over Polygons.

**MNIST:** We use the MNIST dataset (LeCun, 1998) but increase the spatial dimensions to $56 \times 56$ through interpolation. The pixels are binarized at a threshold of 0.5, and are assigned the associated class label of the digit in the image or 'background'. This task is significantly harder than those above since the shapes now differ between instances (i.e. each handwritten 3 is unique) and thus the model must learn these sets of invariances when processing the dynamic representations.

**Multi-MNIST:** Finally, we introduce a variant of the Multi-MNIST dataset (Sabour et al., 2017), where each image contains between 1 and 4 digits placed at random, non-overlapping locations on a 128×128 grid. To construct these images, we first upscale each 28×28 MNIST digit to 42×42 using interpolation, and then position them on the larger canvas. The

pixels are binarized at a threshold of 0.5, with each pixel labeled according to the digit it belongs to or as 'background'. This task is significantly more challenging than previous ones due to the increased spatial dimensions (and thereby greater spatial integration distances), combined with the combinatorial variation in the number and placement of digits. Samples of this new dataset are visualized in Figure 6.

## Models

In the following we detail the local recurrent models and the associated global baselines that we use in this study. For all models, given an input image $\mathbf{x} \in \mathbb{R}^{C \times H \times W}$, the target output is $\mathbf{y} \in \mathbb{R}^{N \times H \times W}$, a set of N class logits for each pixel.

**Locally Coupled Oscillatory RNN (NWM)** As a model which most closely follows the motivational drum analogy introduced earlier, we implement a recurrent neural network parameterized as a network of locally coupled oscillators. In prior work, this model has been referred to as the Neural Wave Machine (NWM) (Keller & Welling, 2023), based on the coRNN (Rusch & Mishra, 2021), and is known to be biased towards traveling wave dynamics. In fact, it is known that in the continuum limit of the number of neurons, such networks of coupled oscillators reduce exactly to wave dynamics of Equation 1 (Schwartz, 2016). Explicitly, the dynamics of this model are given as:

$$\frac{\partial^2 \mathbf{h}}{\partial t^2} = \sigma \left( \mathbf{w}_h \star \mathbf{h} \right) - \gamma_\theta(\mathbf{x}) \odot \mathbf{h} - \alpha_\theta(\mathbf{x}) \odot \frac{\partial \mathbf{h}}{\partial t}. \quad (9)$$

where $\sigma = \tanh$ is the hyperbolic tangent function. In this work, in order to make the recurrent dynamics a function of the input without explicitly clamping hidden states as done in the theoretical section, we modify the original NWM such that the natural frequencies of each oscillator $\gamma$, and the damping term $\alpha$ are a function of the input image, computed through shallow 3-layer CNN models $(\gamma_\theta(\mathbf{x}), \alpha_\theta(\mathbf{x}))$. This crucially allows the way in which waves propagate over the hidden state (and therefore the resulting time-dynamics) to be dictated by the input image, and specifically enables the network to emulate soft boundary conditions via large differences in natural frequencies (as seen in Figure 3). The initial state of the model is also set by the shallow 4-layer CNN encoder $\mathbf{h}_0 = f_\theta(\mathbf{x})$, with $(3 \times 3)$ kernels. Crucially, this yields a receptive field size of $(9 \times 9)$ in the final layer, significantly smaller than the spatial dimensions of all shapes in the datasets listed above (see Figure 7).

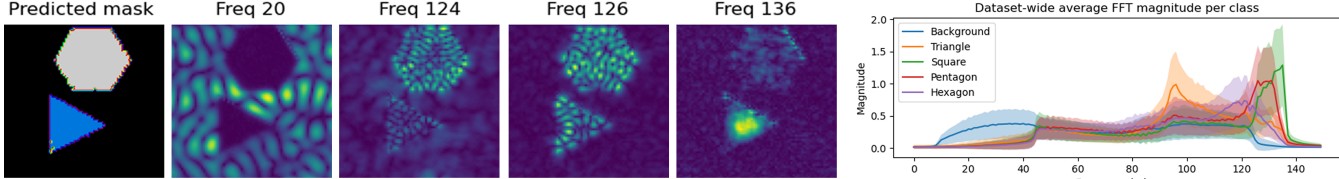

Figure 4: **Wave-based models learn to separate distinct shapes in frequency space.** (Left) Plot of predicted semantic segmentation and a select set of frequency bins for each pixel of a given test image. (Right) The full frequency spectrum for each shape in the dataset, averaged over all pixels containing that class label in the dataset. We see that different shapes have qualitatively different frequency spectra, allowing for $> 99\%$ pixel-wise classification accuracy on a test set.

We initialize the recurrent convolutional kernel $\mathbf{w}_h$ to the finite difference approximation of the Laplacian operator from Equation 8 to bias the model towards wave propagation, and initialize the natural frequency encoder $\gamma_\theta(\mathbf{x})$ to the identity to encourage soft boundaries. Finally, we numerically integrate the second order ODE above using the Implicit-Explicit integration scheme from (Rusch & Mishra, 2021) with a timestep size of $0.1$ for a fixed amount of time (100 timesteps). All recurrent convolutions in this paper are performed with circular padding to avoid boundary effects. We use 2 channels in the recurrent hidden state on MNIST & Tetrominoes, and 16 on Multi-MNIST.

**Convolutional LSTM** To explore how other locally-constrained recurrent architectures may solve these tasks when not explicitly biased towards wave-dynamics *a prioi*, we implement a convolutional variant of the LSTM (Hochreiter & Schmidhuber, 1997), where all previous dense connections are now replaced with local convolutions over the spatial dimensions of the hidden state. Explicitly:

$$\mathbf{x}_0 = f_\theta(\mathbf{x}), \quad \mathbf{i}_t = \sigma\big(\mathbf{w}_{xi} \star \mathbf{x}_{t-1} + \mathbf{w}_{hi} \star \mathbf{h}_{t-1} + \mathbf{b}_i\big), \quad (10)$$

$$\mathbf{f}_t = \sigma\big(\mathbf{w}_{xf} \star \mathbf{x}_{t-1} + \mathbf{w}_{hf} \star \mathbf{h}_{t-1} + \mathbf{b}_f\big), \quad (11)$$

$$\tilde{\mathbf{C}}_t = \tanh\big(\mathbf{w}_{xc} \star \mathbf{x}_{t-1} + \mathbf{w}_{hc} \star \mathbf{h}_{t-1} + \mathbf{b}_c\big), \quad (12)$$

$$\mathbf{C}_t = \mathbf{f}_t \odot \mathbf{C}_{t-1} + \mathbf{i}_t \odot \tilde{\mathbf{C}}_t, \quad \mathbf{h}_t = \mathbf{o}_t \odot \tanh(\mathbf{C}_t), \quad (13)$$

$$\mathbf{o}_t = \sigma\big(\mathbf{w}_{xo} \star \mathbf{x}_{t-1} + \mathbf{w}_{ho} \star \mathbf{h}_{t-1} + \mathbf{b}_o\big), \quad (14)$$

$$\mathbf{x}_t = \mathbf{w}_{o2} \star \sigma(\mathbf{w}_{o1} \star \mathbf{h}_t). \quad (15)$$

$\sigma$ is a sigmoid, and $\odot$ denotes the Hadamard product. As in the NWM, we use two hidden channels, a 4-layer CNN for $f_\theta$, and two 3-layer CNNs to initialize $\mathbf{h}_0$ and $\mathbf{C}_0$. We apply no special initialization, yet as shown in Figure 5, the model still learns to generate hidden-state waves to integrate spatial information and solve the task. All LSTMs are trained with 20 timesteps, which was optimal for small datasets. While 100 timesteps may help in complex tasks, training is substantially slower, so we default to the NWM for larger-scale experiments.

**Recurrent Readout** For the two recurrent models listed above, we must decide how to read out the class label from the sequence of hidden states. The simplest option is to feed the hidden state at the final timestep to a pixel-wise 'readout' network for classification. We denote these models *Last* in Table 1. Alternatively, we can take some function of the hidden

states over a greater extent of the time series, and then feed this output into the readout network. The options for such a time-projection function include taking the maximum or mean hidden state values over time (*Max* and *Mean* respectively), computing the Fourier coefficient amplitudes of the time series (denoted *FFT*), or computing a learnable linear projection of the the full timeseries (denoted *Linear*). In all cases, we parameterize the readout module as a 4-layer MLP for Polygons, MNIST, and Tetrominoes, and a 6-layer MLP for Multi-MNIST.

**Feed-Forward CNN Baselines** The CNN baselines are composed of a number (L) of convolutional layers, each with $(3 \times 3)$ convolutional kernels, and 16 channels. This baseline is intended to demonstrate the inability of neurons with restricted local receptive fields to perform tasks which require global information (for small L), and the ability of global receptive fields to improve this performance (for large L). Explicitly, with $\mathbf{w}^l$ denoting layer $l$'s convolutional kernel and $\sigma$ denoting a *ReLU* (Nair & Hinton, 2010) activation function,

$$\hat{\mathbf{y}} = \sigma(\mathbf{w}^L \star \sigma(\mathbf{w}^{L-1} \star \ldots \sigma(\mathbf{w}^1 \star \mathbf{x} + \mathbf{b}^1) \ldots + \mathbf{b}^{L-1}) + \mathbf{b}^L)), \quad (16)$$

where $\hat{\mathbf{y}}$ is fed channelwise to a linear layer that outputs 100 channels, equivalent to the time-projected output of the RNN models above. The output of this linear layer is then similarly fed into a 'readout' MLP which operates on each pixel's channels individually to produce the class logits.

**Non-local U-Net Baseline** Finally, as a competitive non-local effective upper bound on the local models' performance, we implement a simple U-Net model (Ronneberger et al., 2015) to perform segmentation. Succinctly, these models contain 4 encoder layers that decrease spatial resolution and 4 decoder layers that increase spatial resolution, starting with $c_{in}$ channels, and reaching $c_{in} \cdot 2^4$ channels in the bottleneck. Crucially because of the spatial bottleneck, the receptive fields of pixels in the output layer of this model cover the entire image, allowing for simple solutions to the semantic segmentation task. For the U-Net models, we simply take the final output as logits since the decoder network can be seen as type of 'readout'. We evaluate U-Net on the most challenging dataset: Multi-MNIST.

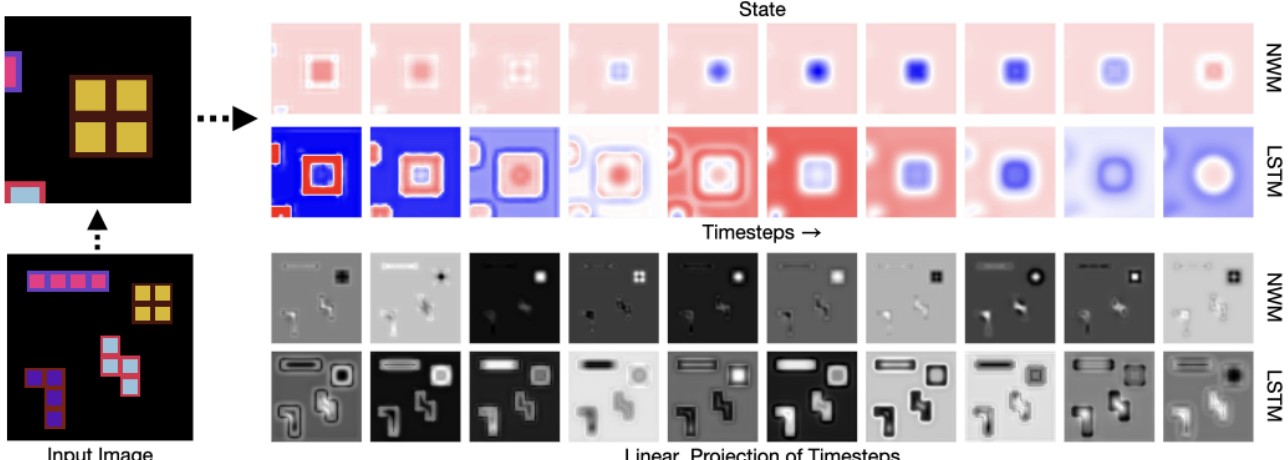

Figure 5: **Both wave-biased and standard local recurrent models learn traveling wave dynamics to integrate spatial information.** Visualization of a subset of the LSTM and NWM hidden state evolution after training (top) for a given image (left). We see waves propagate over the timesteps throughout the shape. On the bottom we plot the associated pixel-wise learned linear projection of the hidden state time-dynamics and observe individual shapes appear to pop out in different learned dimensions.

## Results

**Metrics** In Tables 1 & 2, we present results using standard segmentation metrics: cross-entropy loss ("Loss"), accuracy ("Acc"), and Intersection over Union ("IoU"). IoU measures the ratio of the overlap between the predicted and ground truth regions to the total area covered by both. In the main text, we measure these metrics only on the foreground objects, eliminating the bias introduced by the dominant background in each image. See Supplementary Material for more details.

**Polygons** As an initial proof of concept, in Figure 3 we show the hidden state evolution for the NWM model of Equation 9, with FFT readout, on a single example of the polygons dataset. We see that the model initializes the hidden state such that waves are propagated from the edges of the object in all directions. From naive inspection of the sequence, we can see that waves appear to propagate differently within the object, seemingly changing the spectral representation of each point on the interior of the hexagon. Figure 4 (left) shows the magnitudes of a subset of Fourier coefficients for all neurons in response to a test image, while Figure 4 (right) shows the frequency representations for each object class averaged over all pixels in the validation set which are labeled as that class. We see that there is a clear distinction between the shapes that the model appears to pick up on, which allows it to identify all polygons with $> 99\%$ accuracy on a held out test set. In Appendix Figures 9 & 10, we further quantitatively analyze how the distribution of wavelengths changes during training, and show that this strongly correlates with model performance ($R^2 = 0.961$), strongly implicating wave dynamics in model performance. Additional results are shown in Figures 13 & 14.

**Tetrominoes & MNIST** As a concrete comparison of local recurrent models with different readouts and feed-forward models with different receptive field sizes, in Table 1 we include the aggregated results of 300 total models trained on Tetrominoes and MNIST. We observe that CNNs with 2 and 4 layers perform poorly on both datasets, with performance improving as the receptive field (RF) increases, as expected. This improvement is most pronounced in 16-layer CNNs (with an effective RF size of 33) on MNIST and in 8-layer CNNs (with an effective RF size of 17) on Tetrominoes. However, mean performance declines with 32 layers on MNIST and with 16 and 32 layers on Tetrominoes, though the variance remains high. As deeper networks contain more parameters, they pose a more challenging optimization problem, leading to inconsistent convergence. Nevertheless, peak performance continues to improve as the number of layers increases, even in the 16- and 32-layer models, despite a higher number of failed training runs. For the minimum, maximum, and median performance across different seeds, refer to Tables 5 and 6 in the supplement.

Among recurrent models, those with linear projections perform best, with the NWM outperforming baselines. Recurrent models that rely solely on the last hidden state for predictions achieve the weakest results. The NWM models exhibit the lowest variance, indicating greater training stability. Figure 5 visualizes the recurrent states and linear projections for a sample image. Interestingly, the LSTM learns to generate wave dynamics despite lacking an explicit inductive bias for doing so.

**Multi-MNIST** Most impressively, in Table 2, we see that the NWM with 54K parameters performs better than U-Net's with 30K and 68K parameters, despite having only local connectivity and no explicit skip connections. In addition, our NWM performs only slightly worse than U-Net's with 122K and 190K parameters. Interestingly, the NWM has the lowest foreground loss of all models, possibly due to an increased confidence in predictions (e.g. predicting background) compared with other models. Once again, we see NWM models have much lower variance, suggesting training stability benefits over comparable

U-Net models. These results suggest wave dynamics with linear readouts over time may be a promising avenue to explore as an alternative to U-Net style architectures for integrating spatial information in artificial neural networks.

## Related Work

**Wave-based Computing**   While prior work on wave-based computing in trainable task-oriented neural networks remains scarce, there is a rich history of using wave-like or other spatiotemporal field dynamics generally for computation. Early work studied the ability for waves to perform simple logical operations and thereby compute in a distributed manner (Izhikevich & Hoppensteadt, 2009; Gong & van Leeuwen, 2009), while other work has studied the ability for physical water waves to act as literal instantiations of classic 'reservoir computers' (Maksymov, 2023). Classically, the domain of 'Neural Field Theory' has studied the role of spatiotemporal field dynamics in neural computation from a rigorous mathematical standpoint, although to-date these models have not been adapted to deep-neural network task-oriented performance. We refer readers to Breakspear (2017) for a thorough review of such models.

More recently, Hughes et al. (2019) have noted the analogy between the wave equation and recurrent neural networks, as we have done here, and used this to suggest that wave-based RNNs with learnable wave speeds may perform a type of analog computation. The authors use this to perform acoustic signal classification in a simplified setting, similar to our study in spirit, but differing in how waves are used and their computational purpose. Most related to the present study, Balkenhol et al. (2024) use an architecture similar to ours, with a Laplacian recurrent operator, damping, and gating, to show that when provided with an audio signal at a specific spatial location of the network, neurons at more distant locations can perfectly reconstruct the signal. The authors also show that this network is able to reproduce electrical recordings from macaque monkeys in response to simple grating stimuli, hypothesizing that their detection of high frequency waves is highly related to the transfer of information over large cortical distances.

In terms of task-oriented wave-based models, recent work by Effenberger et al. (2025) extensively studies the computational abilities of oscillatory neural networks, and specifically notes the emergence of traveling waves in these models in response to visual stimuli. Similarly, work by Keller & Welling (2023); Keller et al. (2024b) studies wave-based RNNs for sequence processing and prediction. Our work fundamentally differs from these in the precise study of how these waves may be utilized for the spatial integration of visual information, as is hypothesized to happen in the visual cortex. Furthermore, our work uniquely demonstrates that a timeseries based readout is crucial for performing this type of integration, inspired by Kac's question, opening the door for future novel applications of these models.

**Recurrence vs. Depth**   Another relevant line of research concerns the ability to trade off depth for recurrence in CNNs. Early work in this area was performed by Liao & Poggio (2020),

with a more extensive recent study performed by Schwarzschild et al. (2022). The authors demonstrate how iterating a single convolutional layer in a deep CNN yields similar performance to equivalently deep fully untied CNNs. Our work differs from these in that we demonstrate the advantage of a timeseries readout mechanism, inspired by Kac's question, whereas prior work can be seen as using the 'last' hidden state mechanism, that we see underperforms in this work. Interestingly, our findings thus suggest a potential novel method to improve the performance of these recurrent alternatives to deep networks through the use of our readout, a direction we intend to study in future work. Other more machine learning focused work has studied the impact of various weight-sharing schemes in deep convolutional networks (Eigen et al., 2014; Jastrzębski et al., 2018; Boulch, 2017), however these share the same distinction with the present study in terms of their readout mechanism, while our proposed timeseries readouts appear to be uniquely linked to the wave dynamics that emerge in our models.

**Binding By Synchrony**   Finally, we believe our work shares an interesting connection with the "binding by synchrony" concept (Singer, 2007) from early neuroscience research. Specifically, while our model's 'binding' of parts into wholes does not rely on precise zero-lag synchrony—where oscillators within an object are perfectly in phase, as in the original framework; our method does rely on traveling waves of activity within objects that can be interpreted as a type of phase-lag synchrony. The "binding operation" then involves a transformation of the time signal using a suitable linear projection (our proposed timeseries readout). We believe this connection is valuable precisely since it enables a connection with the extensive historical literature on this concept, while simultaneously forming novel predictions on how such phenomena might manifest in natural neural systems. On the machine learning side of this concept, our work shares a strong connection with a class of object-centric learning methods which leverage a notion of synchrony of neural activations to define 'bound' visual units for computational purposes. This includes models such as complex autoencoders (Löwe et al., 2022, 2024; Stanić et al., 2024; Gopalakrishnan et al., 2024) and recent Artificial Kuramoto Oscillatory Neurons (AKOrN) Miyato et al. (2024). Unlike our method, the waves in the AKOrN model are not used directly as a representation themselves, but instead are neglected through the use of the 'last hidden state' readout method. Perhaps most related to our work, Liboni et al. (2023) use a complex-valued recurrent neural network designed to generate traveling waves for image segmentation, with binding information encoded in the temporal phase sequence of these waves. This method can indeed be seen as using traveling waves to integrate information spatially, but contains no trainable components, offering a more theoretical exposition to the problem, as opposed to the task-oriented empirical study presented here.

## Limitations & Future Work

While we believe the experiments above are convincing of the fact that traveling waves are an effective and efficient mecha-

**MNIST**

| Model | Arch. | #θ | FG-Acc | FG-IoU | FG-Loss |
|---|---|---|---|---|---|
| CNN | 2 | 160k | 0.14 ± 0.06 | 0.10 ± 0.04 | 2.44 ± 0.67 |
|  | 4 | 165k | 0.19 ± 0.11 | 0.13 ± 0.07 | 2.49 ± 0.99 |
|  | 8 | 174k | 0.42 ± 0.15 | 0.30 ± 0.11 | 1.76 ± 0.90 |
|  | **16** | 193k | **0.57 ± 0.39** | **0.50 ± 0.35** | **1.73 ± 1.80** |
|  | 32 | 230k | 0.27 ± 0.43 | 0.25 ± 0.41 | 3.12 ± 1.91 |
| LSTM | Max | 146k | 0.42 ± 0.05 | 0.31 ± 0.04 | 1.57 ± 0.13 |
|  | Mean | 146k | 0.42 ± 0.09 | 0.31 ± 0.07 | 1.57 ± 0.22 |
|  | Last | 146k | 0.32 ± 0.23 | 0.24 ± 0.18 | 2.34 ± 1.38 |
|  | FFT | 151k | 0.73 ± 0.27 | 0.66 ± 0.25 | 0.93 ± 1.23 |
|  | **Linear** | 151k | **0.79 ± 0.28** | **0.73 ± 0.26** | **0.78 ± 1.25** |
| **NWM** | Max | 151k | 0.48 ± 0.07 | 0.37 ± 0.07 | 1.39 ± 0.17 |
|  | Mean | 151k | 0.46 ± 0.08 | 0.35 ± 0.07 | 1.47 ± 0.21 |
|  | Last | 151k | 0.50 ± 0.08 | 0.39 ± 0.07 | 1.34 ± 0.19 |
|  | FFT | 177k | 0.76 ± 0.05 | 0.65 ± 0.07 | 0.72 ± 0.15 |
|  | **Linear** | 177k | **0.90 ± 0.02** | **0.85 ± 0.04** | **0.30 ± 0.07** |

**Tetrominoes**

| Model | Arch. | #θ | FG-Acc | FG-IoU | FG-Loss |
|---|---|---|---|---|---|
| CNN | 2 | 160k | 0.24 ± 0.08 | 0.14 ± 0.05 | 1.74 ± 0.62 |
|  | 4 | 164k | 0.31 ± 0.16 | 0.20 ± 0.11 | 1.76 ± 0.94 |
|  | **8** | 173k | **0.74 ± 0.26** | **0.64 ± 0.22** | **0.70 ± 1.00** |
|  | 16 | 192k | 0.40 ± 0.51 | 0.40 ± 0.51 | 2.14 ± 1.83 |
|  | 32 | 229k | 0.33 ± 0.47 | 0.32 ± 0.47 | 2.26 ± 1.67 |
| LSTM | Max | 146k | 0.62 ± 0.30 | 0.52 ± 0.31 | 0.99 ± 0.95 |
|  | Mean | 146k | 0.64 ± 0.25 | 0.53 ± 0.23 | 0.95 ± 0.96 |
|  | Last | 146k | 0.59 ± 0.41 | 0.52 ± 0.38 | 1.37 ± 1.51 |
|  | FFT | 151k | 0.95 ± 0.04 | 0.91 ± 0.07 | 0.18 ± 0.11 |
|  | **Linear** | 151k | **0.97 ± 0.02** | **0.94 ± 0.03** | **0.13 ± 0.05** |
| **NWM** | Max | 151k | 0.88 ± 0.11 | 0.81 ± 0.15 | 0.35 ± 0.24 |
|  | Mean | 151k | 0.92 ± 0.07 | 0.87 ± 0.11 | 0.24 ± 0.14 |
|  | Last | 151k | 0.94 ± 0.06 | 0.90 ± 0.09 | 0.21 ± 0.14 |
|  | FFT | 176k | 0.98 ± 0.01 | 0.97 ± 0.01 | 0.07 ± 0.03 |
|  | **Linear** | 176k | **0.99 ± 0.00** | **0.98 ± 0.01** | **0.05 ± 0.02** |

Table 1: **Locally constrained recurrent models with timeseries readouts can semantically segment images at the pixel level, a task requiring global information.** Models with the lowest foreground loss are in bold. 'Arch' refers to # of CNN layers, and type of readout for LSTM and NWM. #θ refers to the number of parameters. Results are from 10 random seeds ($mean \pm std$).

| Model | #θ | FG-Acc | FG-IoU | FG-Loss |
|---|---|---|---|---|
| U-Net 2c | 31K | 0.66 ± 0.27 | 0.60 ± 0.28 | 1.19 ± 0.56 |
| U-Net 3c | 69K | 0.90 ± 0.10 | 0.87 ± 0.13 | 0.56 ± 0.26 |
| NWM | 55K | 0.96 ± 0.01 | 0.93 ± 0.01 | 0.15 ± 0.02 |
| U-Net 4c | 122K | 0.97 ± 0.00 | 0.96 ± 0.01 | 0.24 ± 0.04 |
| U-Net 5c | 190K | 0.98 ± 0.00 | 0.97 ± 0.00 | 0.17 ± 0.05 |

Table 2: **Wave-based models outperform comparably sized U-Net models on more challenging Multi-MNIST segmentation.** U-Net #c refers to the number of feature maps output by the first layer, doubling each layer thereafter, and #θ refers to the number of parameters. Rows sorted by #θ. Results are from 12 random seeds, displayed as $mean \pm std$.

nism for integrating spatial information through the time dimensions, they are inherently limited in a number of ways. First, from the machine learning perspective, while the wave-based models may be able to outperform U-Net models with an equivalent number of parameters on the Multi-MNIST task presented here, the amount of computation time is significantly higher to run these models on our current hardware. This is due to the fact that the oscillatory wave dynamics must be accurately numerically integrated (with a small $\Delta t$), while U-Net type models are optimized for parallel GPU hardware. In future work, we intend to explore the potential of using oscillatory state space models (Rusch & Rus, 2025) to enable the parallel processing of the recurrent NWM dynamics over sequence length, which would significantly lessen this computational bottleneck. From the neuroscience perspective, our proposed wave-based models are highly abstract idealizations of a cortical sheet, thereby allowing for the tractable computation, but also obscuring how some of the parameters such as the natural frequencies or damping parameters could be mapped onto neurobiological components. Despite this limitation, this remains one of the few models which can be trained to leverage wave dynamics in a task-oriented manner; and therefore, in future work, we intend to leverage this uniquely new framework to compare the learned dynamics with neural recordings in a precise manner. Finally, although our work is inspired by the 'hearing the shape of a drum' problem, we have no guarantee that our trained models form image representations in precisely that manner. We do find the analogy valuable for guiding model development, and the success of time-based readouts supports this intuition. However, we caution readers against overinterpreting this analogy as a literal account of how the models operate.

## Conclusion

In the above, we have presented arguments both theoretical and empirical supporting the idea that traveling waves may serve to integrate spatial information through the time dimension in otherwise locally constrained architectures, achieving performance comparable with globally connected counterparts. Furthermore, we have demonstrated empirically that this wave-encoded information is most directly accessible through linear projections of the hidden state time-dynamics, contrary to how most recurrent alternatives to depth have previously been studied. We showed that even if models are not biased towards wave dynamics initially, such as the Conv-LSTM, they will still learn to propagate waves in order to transfer information effectively through space, thereby implicating waves and wave-based representations as an optimal solution to information transfer under such constraints. Finally, we demonstrated that wave-based integration of information may be a stable

and parameter efficient rival to common U-Net architectures. Notably, these wave-based solutions—naturally spanning both spatial and frequency domains—could align more directly with EEG or MEG measurements in neuroscience, while on the machine learning side we speculate they could someday help alleviate the computational bottlenecks of global self-attention mechanisms. We hope that this work draws increased attention to the idea that wave-based representations may carry global task-relevant information in both biological and artificial systems, thereby encouraging their further study.

## Acknowledgments

This work has been made possible in part by a gift from the Chan Zuckerberg Initiative Foundation to establish the Kempner Institute for the Study of Natural and Artificial Intelligence at Harvard University. MJ is supported by the Kempner Institute Graduate Research Fellowship. TAK is supported by the Kempner Institute Research Fellowship. Additionally, we would like to thank members of the CRISP Lab at Harvard, and the Kempner Institute for the opportunity to present and improve early versions of this work.

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

## Supplementary Material

### Experimental Details

This section provides details on the training and evaluation procedures for the models presented in this paper. The full code for reproducing results and visualizations from the main text is available at: `https://github.com/KempnerInstitute/traveling-waves-integrate`.

Each model is trained for 300 epochs on the MNIST, Tetrominoes, and Multi-MNIST datasets. We evaluate the validation loss at the end of every epoch and retain the model with the lowest validation loss throughout training. The training process employs the Adam optimizer (Kingma & Ba, 2017) with a learning rate of 0.001 and a batch size of 64.

For dataset partitioning, we use 51,000 images for training, 9,000 for validation, and 10,000 for testing in MNIST. The Tetrominoes dataset consists of 10,000 images for training, 1,000 for validation, and 1,000 for testing. Similarly, the Multi-MNIST dataset comprises 10,000 images for training, 1,000 for validation, and 1,000 for testing. Table 3 reports pixel-wise accuracy, IoU, and loss for both foreground and background.

Each model is trained using multiple random seeds. For MNIST and Tetrominoes, we train each model using 10 different random seeds, while for Multi-MNIST, we use 12 seeds. For example, the NWM with a linear readout is trained on MNIST 10 times, each with a different random seed. After training, we evaluate each model individually and present the aggregated results, including the mean and standard deviation, in Tables 1, 2, and 3. In total, we train 150 models on MNIST, 150 models on Tetrominoes, and 60 models on Multi-MNIST, leading to a total of 360 models.

We train the Conv-LSTMs for 20 timesteps. The NWM model is trained for 100 timesteps on the MNIST, Tetrominoes, and Multi-MNIST datasets, while for the polygons dataset, the NWM runs for 500 timesteps. In the FFT readout, we use the real component of the discrete Fourier transform. This results in 50 bins for the NWM on MNIST, Tetrominoes, and Multi-MNIST, and 250 bins for the polygons dataset. The LSTM model outputs 10 Fourier bins for MNIST, Tetrominoes, and Multi-MNIST.

The Conv-LSTMs and NWMs use nearly identical CNN encoders. Both use the same 4-layer CNN to process the input $x$ (to initialize the input $x_0$ for the LSTM and to initialize the hidden state $h_0$ for the NWM). The LSTM uses two 3-layer CNNs to intialize the hidden and cell states. The NWM uses two identical (to the LSTM) 3-layer CNNs to initialize the natural frequencies $\gamma$ and damping term $\alpha$, except the NWM 3-layer CNN's include an extra $ReLU$ function at the end of the CNN to support waves.

All convolutional models are trained with 16 channels. To ensure a fair comparison with the NWM, a linear layer operating channelwise outputs 100 channels. The readout MLP used for MNIST and Tetrominoes consists of four layers. Its input size is given by the number of Fourier bins multiplied by 2, followed by two hidden layers of 256 neurons each, with ReLU activation between layers, and a final output layer producing logits for classification. On Multi-MNIST, the NWM employs a six-layer readout with 32 neurons in each hidden layer.

For the U-Net architecture, the Arch parameter in Tables 2 and 3 refers to the number of feature maps output by the first convolutional layer. Each U-Net begins with stacked convolutions that maintain spatial resolution, producing $Arch$ feature maps (e.g., 3). The following four layers apply multiple convolutional operations per layer, each reducing spatial resolution by half while doubling the number of feature maps. If the initial layer outputs 3 feature maps, the subsequent layers modify the number as follows:

$$3 \rightarrow 6 \rightarrow 12 \rightarrow 24 \rightarrow 48.$$

The final number of feature maps, determined by $Arch$, follows:

$$2 \rightarrow 32, \quad 3 \rightarrow 48, \quad 4 \rightarrow 64, \quad 5 \rightarrow 80.$$

The U-Net decoder progressively upsamples the spatial resolution over four layers while simultaneously reducing the number of feature maps by half. By the fourth layer, it restores the feature map count to the value specified by $Arch$. Finally, 1×1 convolutions are used to project the feature maps into logits for pixel-wise classification.

## Metrics

In the main text, we present results using pixel-wise cross-entropy loss, accuracy, and IoU. Suppose we have N classes. Below, we define exactly how each is computed. Suppose we have predicted segmentation mask $\hat{y} \in \{0, 1, ..., N\}^{H \times W}$, ground truth segmentation mask $y \in \{0, 1, ..., N\}^{H \times W}$, and foreground mask $M \in \{0, 1\}^{H \times W}$ (where $M_{ij} = 1$ if and only if $X_{ij}$ is a foreground pixel, i.e. $y_{ij} \neq 0$). We also have $FG\_count = \sum_{i=1}^{H} \sum_{j=1}^{W} M_{ij}$, i.e. the number of foreground pixels. For a given image, suppose we have $\hat{l} \in \mathbb{R}^{H \times W \times N}$, where $\hat{l}_{ij}$ is the predicted probability of each of the N classes, such that $\hat{y}_{ij} = \text{argmax}_n(\hat{l}_{ij})$.

$$\text{Cross-Entropy}(\hat{l}, l) = -\frac{1}{HW} \sum_{i=1}^{H} \sum_{j=1}^{W} \sum_{n=1}^{N} \mathbf{1}_{\{y_{ij}=n\}} \log(\hat{l}_{ijn}). \tag{17}$$

$$\text{Accuracy}(\hat{y}, y) = \frac{1}{HW} \sum_{i=1}^{H} \sum_{j=1}^{W} \mathbf{1}_{\{\hat{y}_{ij}=y_{ij}\}}. \tag{18}$$

$$\text{IoU}(\hat{y}, y) = \frac{1}{HW} \sum_{i=1}^{H} \sum_{j=1}^{W} \frac{\text{Intersection}(\hat{y}_{ij}, y_{ij})}{\text{Union}(\hat{y}_{ij}, y_{ij})} \tag{}$$

$$\text{FG-Cross-Entropy}(\hat{l}, y) = -\frac{1}{FG\_count} \sum_{i=1}^{H} \sum_{j=1}^{W} M_{ij} \sum_{n=1}^{N} \mathbf{1}_{\{y_{ij}=n\}} \log(\hat{l}_{ijn}). \tag{19}$$

$$\text{FG-Accuracy}(\hat{y}, y) = \frac{1}{FG\_count} \sum_{i=1}^{H} \sum_{j=1}^{W} M_{ij} \mathbf{1}_{\{\hat{y}_{ij}=y_{ij}\}}, \tag{20}$$

$$\text{FG-IoU}(\hat{y}, y) = \frac{1}{FG\_count} \sum_{i=1}^{H} \sum_{j=1}^{W} M_{ij} \times \frac{\text{Intersection}(\hat{y}_{ij}, y_{ij})}{\text{Union}(\hat{y}_{ij}, y_{ij})}. \tag{21}$$

Where $\text{Intersection}(\hat{y}_{ij}, y_{ij}) = 1$ if $\hat{y}_{ij} = y_{ij}$ and 0 otherwise; and $\text{Union}(\hat{y}_{ij}, y_{ij}) = 1$ if $\hat{y}_{ij} = y_{ij}$ and 2 otherwise.

## Multi-MNIST Dataset

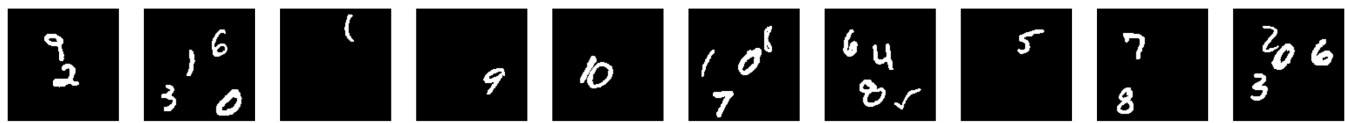

Figure 6: Samples from the newly generated Multi-MNIST dataset.

**Example of Receptive Field**

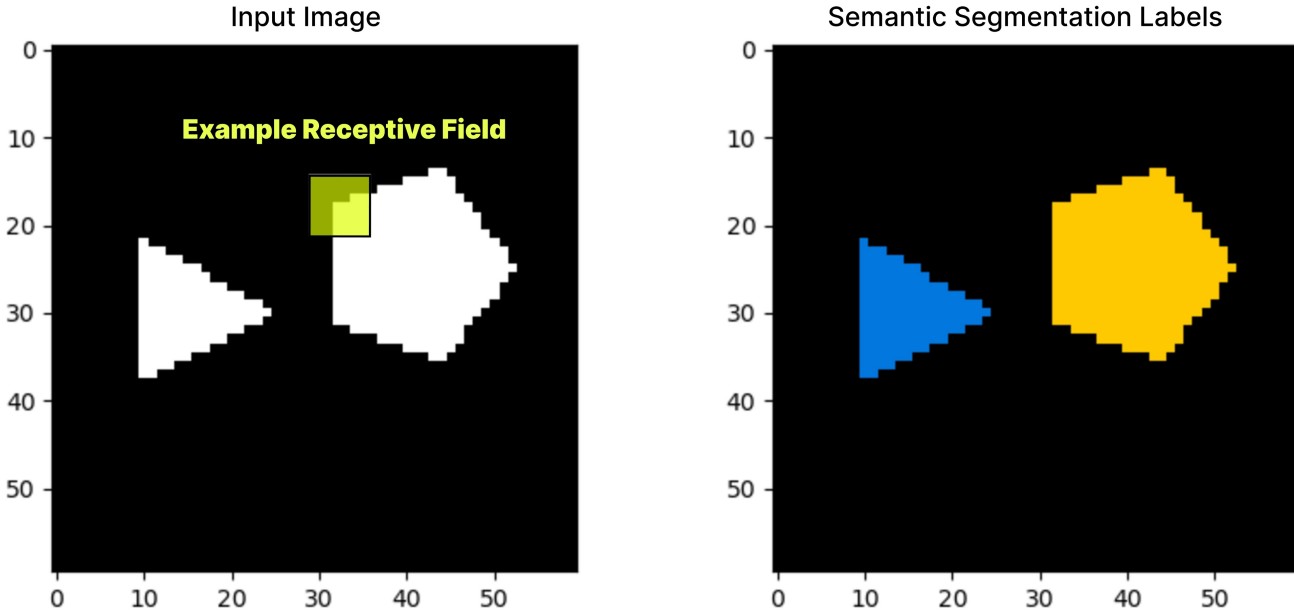

Figure 7: Example of receptive field overlayed with objects. We see that the dataset and model architectures have been intentionally designed such that the receptive field size is smaller than the size required to fully capture the shape information necessary to classify the shape.

**NWM Recurrent Kernel Before and After Training**

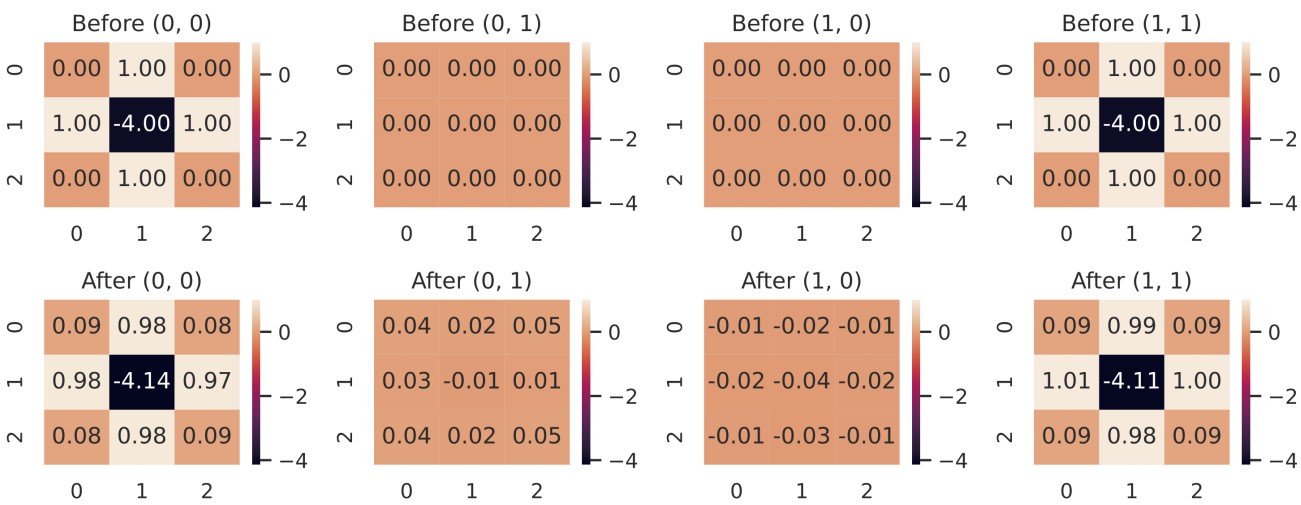

Figure 8: NWM recurrent kernel weights before (top) and after (bottom) training on Tetrominoes. We see that after training, the weights have not significantly deviated from their Laplacian initialization, maintaining wave-like recurrent dynamics.

## Extended Quantitative Analysis of Wave Dynamics

In order to demonstrate the importance of wave dynamics in the trained models presented in the main text, in this section we include additional quantitative analysis of the emergence of wave dynamics during training and the correlation with model performance.

First, to compute a metric for the degree of wave propagation during training, we leverage a method from the neuroscience literature (Davis et al., 2020, 2021) which computes an estimated instantaneous wavelength at each spatial location. In prior work, the detection of long wavelengths above baseline length has proven an accurate measure of the existence of structured wave-like dynamics in otherwise highly noisy dynamics.

In detail, the method transforms the initial real-valued signal $\mathbf{x}(t)$ (e.g. the hidden state of our RNNs) to a corresponding analytic complex valued signal $\mathbf{x}_a(t)$ through the Hilbert Transform $\mathcal{H}$. Explicitly: $\mathbf{x}_a(t) = \mathbf{x}(t) + i\mathcal{H}[\mathbf{x}(t)]$. The instantaneous phase of the signal can then be computed at each point in space through the complex argument of this analytic signal: $\phi(t) = \mathrm{Arg}[\mathbf{x}_a(t)]$. The wavelength at each location is then given simply as the spatial gradient of this phase: $\nu(t) = -\nabla\phi(t)$.

In Figure 9, we plot the distribution of these estimated wavelengths over training iterations for an NWM model trained on the polygons dataset, and observe that there is a significant trend towards the development of longer wavelengths as training progresses. In otherwords, the wavelength distribution gets significantly more *heavy-tailed* through training.

To get a single summary statistic to quantify the development of this distribution, we compute the percentage of spatial locations where the wavelength exceeds a minimal threshold (in this case we pick a threshold of 1.5, large enough to ignore the peak at 1 due to discretization, but we find similar results for a wide range of thresholds). In Figure 10, we plot this wavelength percentage for each training iteration, along with the corresponding (negative) loss of the model at the same iteration. Indeed, we see that these two values are highly correlated with the loss decreasing at the same rate as long wavelengths emerge. Quantitatively, these two values, averaged over 5 random seeds, reach a correlation coefficient of 0.961.

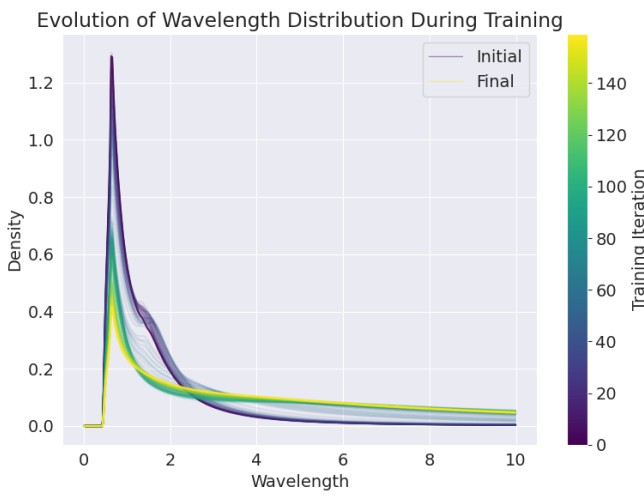
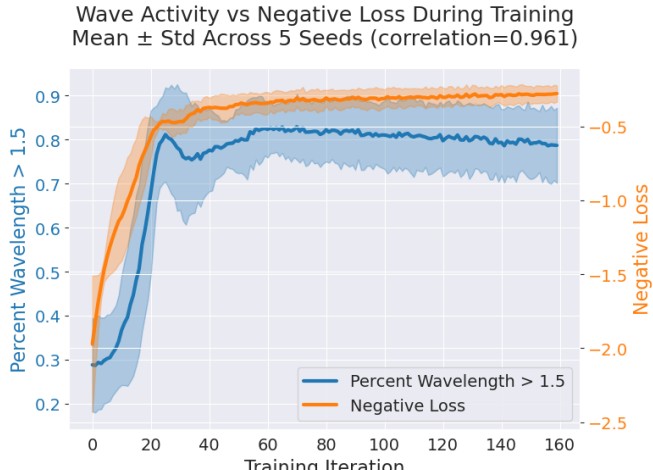

Figure 9: Distribution of estimated wavelengths over training for the NWM model trained on the polygons dataset. We see that, with training, significantly more structured dynamics emerge, evidenced by longer estimated wavelengths, i.e. the distribution gets significantly more heavy-tailed with training.

Figure 10: Comparison of the emergence of long-wavelengths detected in the NWM neural dynamics, and the associated loss of the model over training. We see that there is a strong correlation between the emergence of waves and the improved performance of the model.

## Extended Visualizations of Table 1

Below, we include visualizations of the MNIST and Tetrominoes results from Table 1 in an alternative format to highlight our key findings. In Figure 11, we analyze the effect of the readout mechanism for both the MNIST dataset (top 6 plots) and the Tetronimoes dataset (bottom 6 plots). We include a dashed red line in the Acc / IoU metrics to indicate the performance of a baseline predictor which only predicts background for each pixel. For the FG-ACC and FG-IoU metrics, this predictor would achieve 0.0, and is thus omitted.

We see that the NWM and LSTM perform comparably on all metrics across both datasets. Most interestingly, we see that the linear readout mechanism performs the best for both models, followed by the FFT. We additionally see the results on Tetronimoes are significantly less varied than on MNIST, likely owing to the increased simplicity of the shapes int he dataset.

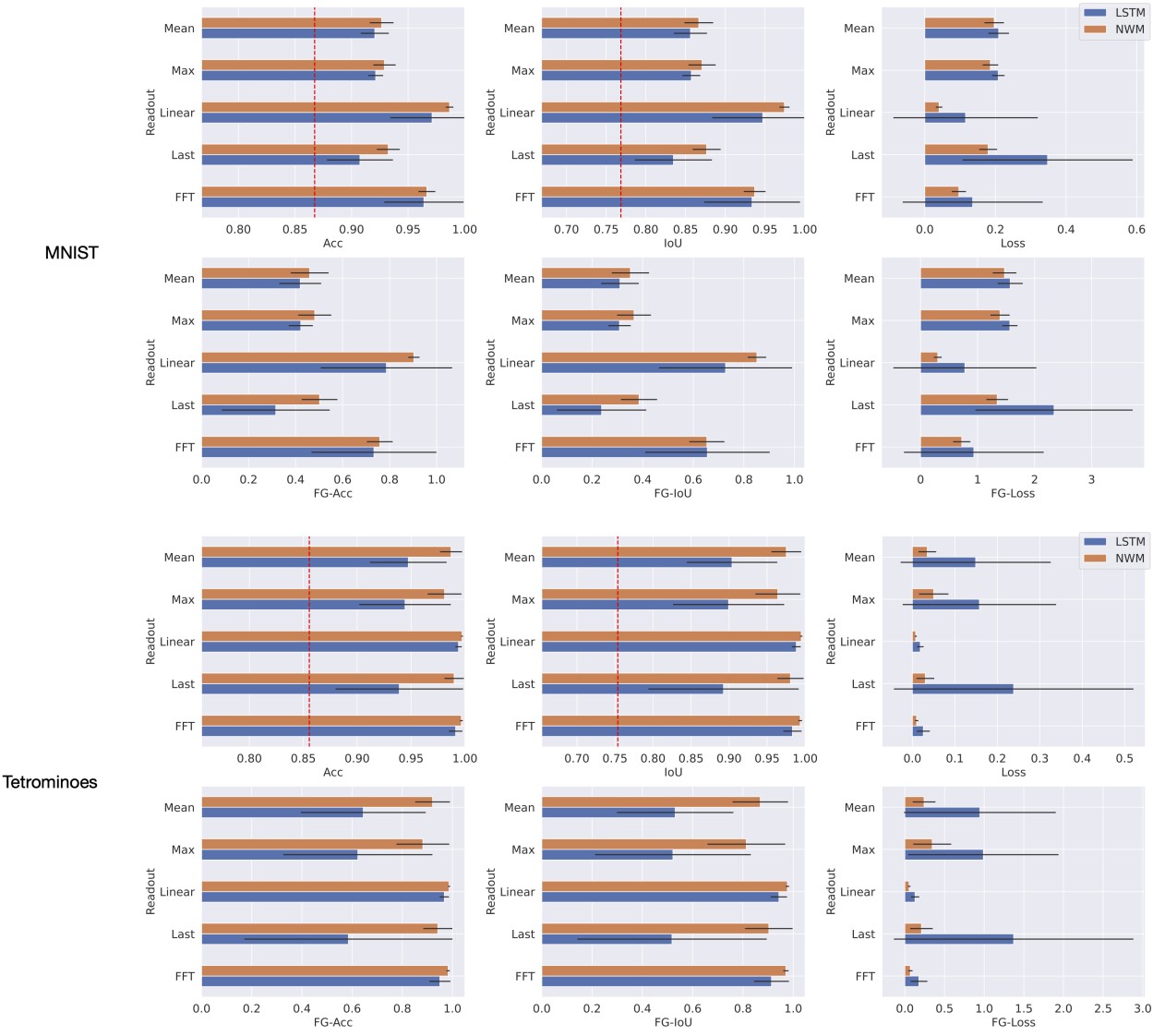

Figure 11: Comparing readout types on MNIST and Tetrominoes. Dashed red line indicates performance of a (baseline) predictor that predicts the background for every pixel.

In Figure 12, we visualize the performance of the recurrent neural networks with the best identified readout mechanism from the prior plot (linear) and compare this to the performance of CNN models of varying depths on both the MNIST dataset (top 6 plots) and the Tetronimoes dataset (bottom 6 plots). We again include a dashed red line in the Acc / IoU metrics to indicate the performance of a baseline predictor which only predicts background for each pixel. We see that the CNNs have significantly higher variance and thus consistently underperform the recurrent alternatives on average.

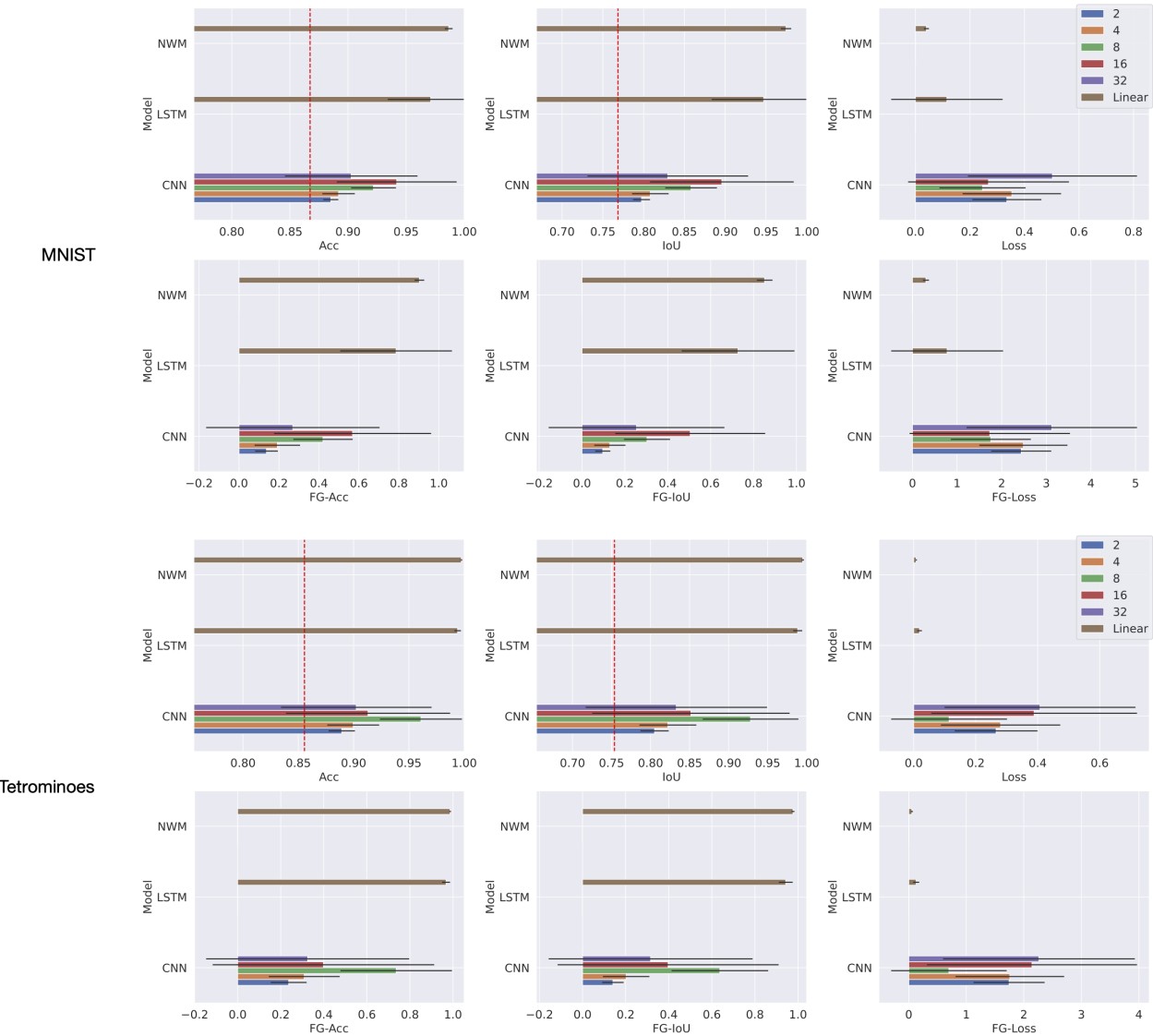

Figure 12: Comparing depth (CNNs) vs recurrence (LSTM and NWM) on MNIST and Tetrominoes. Dashed red line indicates performance of a (baseline) predictor that predicts the background for every pixel.

**Extended Semantic Segmentation Tabular Results**

In Tables 3 and 4 we include the accuracies, IoU, and Loss values computed over the full set of classes (including the background class) for all models presented in the main text. While these numbers are artificially inflated from the inclusion of the background class (which dominates the majority of pixels) we include them here for completeness.

In Tables 5 and 6, we include the minimum, maximum, and median accuracies, IoU, and loss values for MNIST and Tetrominoes to provide a better intuition for the full distribution of each model's performance over different random seeds. We see that the variance implied by these values is in-line with the standard deviations reported in the main text.

| Dataset | Model | Architecture | Acc | IOU | Loss |
|---|---|---|---|---|---|
| MNIST | CNN | 2 | 0.89 ± 0.01 | 0.80 ± 0.01 | 0.34 ± 0.13 |
| | | 4 | 0.89 ± 0.01 | 0.81 ± 0.02 | 0.35 ± 0.18 |
| | | 8 | 0.92 ± 0.02 | 0.86 ± 0.03 | 0.25 ± 0.16 |
| | | **16** | **0.94 ± 0.05** | **0.90 ± 0.09** | **0.27 ± 0.29** |
| | | 32 | 0.90 ± 0.06 | 0.83 ± 0.10 | 0.50 ± 0.31 |
| | LSTM | Max | 0.92 ± 0.01 | 0.86 ± 0.01 | 0.21 ± 0.02 |
| | | Mean | 0.92 ± 0.01 | 0.86 ± 0.02 | 0.21 ± 0.03 |
| | | Last | 0.91 ± 0.03 | 0.83 ± 0.05 | 0.35 ± 0.24 |
| | | FFT | 0.96 ± 0.04 | 0.93 ± 0.06 | 0.14 ± 0.20 |
| | | **Linear** | **0.97 ± 0.04** | **0.95 ± 0.06** | **0.12 ± 0.20** |
| | **NWM** | Max | 0.93 ± 0.01 | 0.87 ± 0.02 | 0.19 ± 0.02 |
| | | Mean | 0.93 ± 0.01 | 0.87 ± 0.02 | 0.20 ± 0.03 |
| | | Last | 0.93 ± 0.01 | 0.88 ± 0.02 | 0.18 ± 0.03 |
| | | FFT | 0.97 ± 0.01 | 0.94 ± 0.01 | 0.10 ± 0.02 |
| | | **Linear** | **0.99 ± 0.00** | **0.97 ± 0.01** | **0.04 ± 0.01** |
| Tetrominoes | CNN | 2 | 0.89 ± 0.01 | 0.81 ± 0.02 | 0.27 ± 0.13 |
| | | 4 | 0.90 ± 0.02 | 0.82 ± 0.04 | 0.28 ± 0.19 |
| | | **8** | **0.96 ± 0.04** | **0.93 ± 0.06** | **0.11 ± 0.19** |
| | | 16 | 0.91 ± 0.07 | 0.85 ± 0.13 | 0.39 ± 0.33 |
| | | 32 | 0.90 ± 0.07 | 0.83 ± 0.12 | 0.41 ± 0.31 |
| | LSTM | Max | 0.94 ± 0.04 | 0.90 ± 0.07 | 0.16 ± 0.18 |
| | | Mean | 0.95 ± 0.04 | 0.90 ± 0.06 | 0.15 ± 0.18 |
| | | Last | 0.94 ± 0.06 | 0.89 ± 0.10 | 0.24 ± 0.28 |
| | | FFT | 0.99 ± 0.01 | 0.98 ± 0.01 | 0.03 ± 0.02 |
| | | **Linear** | **0.99 ± 0.00** | **0.99 ± 0.01** | **0.02 ± 0.01** |
| | **NWM** | Max | 0.98 ± 0.02 | 0.96 ± 0.03 | 0.05 ± 0.03 |
| | | Mean | 0.99 ± 0.01 | 0.98 ± 0.02 | 0.04 ± 0.02 |
| | | Last | 0.99 ± 0.01 | 0.98 ± 0.02 | 0.03 ± 0.02 |
| | | FFT | 1.00 ± 0.00 | 0.99 ± 0.00 | 0.01 ± 0.00 |
| | | **Linear** | **1.00 ± 0.00** | **0.99 ± 0.00** | **0.01 ± 0.00** |

Table 3: Supervised segmentation performance of various models and spectral methods. Models with the lowest foreground loss are in bold. Arch (architecture) for the CNN refers to the number of layers, while for the LSTM and NWM refers to the type of recurrent readout used. Each model is trained with 10 random seeds, and the results are displayed as $mean \pm standard\ deviation$ over the 10 seeds.

| Model | Arch. | Parameters | Acc | IoU | Loss |
|---|---|---|---|---|---|
| U-Net | 2 | 30745 | 0.98 ± 0.01 | 0.97 ± 0.02 | 0.06 ± 0.03 |
| | 3 | 68834 | 1.00 ± 0.00 | 0.99 ± 0.01 | 0.03 ± 0.01 |
| | 4 | 122071 | 1.00 ± 0.00 | 1.00 ± 0.00 | 0.01 ± 0.00 |
| | 5 | 190456 | 1.00 ± 0.00 | 1.00 ± 0.00 | 0.01 ± 0.00 |
| NWM | Linear | 54855 | 1.00 ± 0.00 | 1.00 ± 0.00 | 0.01 ± 0.00 |

Table 4: Supervised segmentation performance of UNet and NWM with Linear Time Projection on Multi-MNIST. Arch for the U-Net refers to the number of feature maps output by the first layer. The number of feature maps doubles between each layer (e.g. 3 means $3 \rightarrow 6 \rightarrow 12 \rightarrow 24 \rightarrow 48$ by the final layer). For the NWM, Arch (architecture) refers to the type of recurrent readout used. Each model is trained with 12 random seeds, and the results are displayed as $mean \pm standard\ deviation$ over the 12 seeds.

| | | | Acc | IoU | Loss |
|---|---|---|---|---|---|
| MNIST | CNN | 2 | 0.89 / 0.89 / 0.87 | 0.80 / 0.80 / 0.77 | 0.70 / 0.29 / 0.29 |
| | | 4 | 0.90 / 0.90 / 0.87 | 0.83 / 0.82 / 0.77 | 0.70 / 0.27 / 0.26 |
| | | 8 | 0.93 / 0.93 / 0.87 | 0.87 / 0.87 / 0.77 | 0.70 / 0.20 / 0.19 |
| | | 16 | 0.98 / 0.97 / 0.87 | 0.96 / 0.95 / 0.77 | 0.70 / 0.09 / 0.07 |
| | | 32 | 0.99 / 0.87 / 0.87 | 0.98 / 0.77 / 0.77 | 0.70 / 0.69 / 0.05 |
| | LSTM | Linear | 0.99 / 0.98 / 0.87 | 0.98 / 0.97 / 0.77 | 0.70 / 0.05 / 0.03 |
| | | FFT | 0.98 / 0.97 / 0.87 | 0.97 / 0.95 / 0.77 | 0.70 / 0.07 / 0.05 |
| | | Last | 0.94 / 0.92 / 0.87 | 0.89 / 0.85 / 0.77 | 0.70 / 0.21 / 0.16 |
| | | Mean | 0.94 / 0.92 / 0.90 | 0.88 / 0.86 / 0.82 | 0.26 / 0.20 / 0.17 |
| | | Max | 0.93 / 0.92 / 0.91 | 0.87 / 0.86 / 0.84 | 0.24 / 0.20 / 0.19 |
| | NWM | Linear | 0.99 / 0.99 / 0.98 | 0.98 / 0.97 / 0.96 | 0.06 / 0.04 / 0.03 |
| | | FFT | 0.98 / 0.97 / 0.95 | 0.96 / 0.94 / 0.91 | 0.14 / 0.09 / 0.07 |
| | | Last | 0.95 / 0.94 / 0.91 | 0.90 / 0.88 / 0.84 | 0.23 / 0.17 / 0.14 |
| | | Mean | 0.94 / 0.93 / 0.91 | 0.89 / 0.87 / 0.84 | 0.24 / 0.20 / 0.16 |
| | | Max | 0.94 / 0.93 / 0.92 | 0.90 / 0.87 / 0.85 | 0.21 / 0.20 / 0.15 |
| Tetrominoes | CNN | 2 | 0.89 / 0.89 / 0.86 | 0.81 / 0.81 / 0.75 | 0.65 / 0.22 / 0.22 |
| | | 4 | 0.91 / 0.91 / 0.86 | 0.84 / 0.84 / 0.75 | 0.65 / 0.19 / 0.18 |
| | | 8 | 0.97 / 0.97 / 0.86 | 0.95 / 0.95 / 0.75 | 0.65 / 0.06 / 0.05 |
| | | 16 | 1.00 / 0.86 / 0.86 | 1.00 / 0.75 / 0.75 | 0.65 / 0.65 / 0.00 |
| | | 32 | 1.00 / 0.86 / 0.86 | 1.00 / 0.75 / 0.75 | 0.65 / 0.63 / 0.00 |
| | LSTM | Linear | 1.00 / 1.00 / 0.99 | 0.99 / 0.99 / 0.97 | 0.04 / 0.02 / 0.01 |
| | | FFT | 1.00 / 0.99 / 0.98 | 0.99 / 0.99 / 0.95 | 0.06 / 0.02 / 0.01 |
| | | Last | 0.99 / 0.98 / 0.86 | 0.97 / 0.95 / 0.75 | 0.65 / 0.06 / 0.04 |
| | | Mean | 0.99 / 0.95 / 0.86 | 0.99 / 0.91 / 0.75 | 0.65 / 0.10 / 0.03 |
| | | Max | 0.99 / 0.94 / 0.86 | 0.98 / 0.88 / 0.75 | 0.65 / 0.13 / 0.03 |
| | NWM | Linear | 1.00 / 1.00 / 1.00 | 1.00 / 0.99 / 0.99 | 0.01 / 0.01 / 0.00 |
| | | FFT | 1.00 / 1.00 / 0.99 | 1.00 / 0.99 / 0.99 | 0.02 / 0.01 / 0.01 |
| | | Last | 1.00 / 0.99 / 0.97 | 0.99 / 0.99 / 0.94 | 0.08 / 0.02 / 0.01 |
| | | Mean | 1.00 / 0.99 / 0.97 | 1.00 / 0.98 / 0.94 | 0.07 / 0.03 / 0.01 |
| | | Max | 1.00 / 0.99 / 0.95 | 0.99 / 0.97 / 0.91 | 0.12 / 0.04 / 0.01 |

Table 5: Segmentation performance (only Acc, IoU, and Loss). Values are max / median / min over 10 seeds.

|  |  |  | FG-Acc | FG-IoU | FG-Loss |
|---|---|---|---|---|---|
| MNIST | CNN | 2 | 0.18 / 0.17 / 0.00 | 0.12 / 0.10 / 0.00 | 4.35 / 2.21 / 2.20 |
|  |  | 4 | 0.28 / 0.25 / 0.00 | 0.18 / 0.17 / 0.00 | 4.37 / 2.00 / 1.93 |
|  |  | 8 | 0.49 / 0.47 / 0.00 | 0.36 / 0.33 / 0.00 | 4.31 / 1.47 / 1.42 |
|  |  | 16 | 0.83 / 0.81 / 0.00 | 0.75 / 0.72 / 0.00 | 4.35 / 0.63 / 0.55 |
|  |  | 32 | 0.91 / 0.00 / 0.00 | 0.86 / 0.00 / 0.00 | 4.37 / 4.29 / 0.33 |
|  | LSTM | Linear | 0.93 / 0.88 / 0.00 | 0.88 / 0.82 / 0.00 | 4.33 / 0.35 / 0.24 |
|  |  | FFT | 0.89 / 0.81 / 0.00 | 0.83 / 0.72 / 0.00 | 4.39 / 0.56 / 0.35 |
|  |  | Last | 0.58 / 0.40 / 0.00 | 0.46 / 0.28 / 0.00 | 4.34 / 1.58 / 1.18 |
|  |  | Mean | 0.53 / 0.43 / 0.27 | 0.41 / 0.32 / 0.20 | 1.97 / 1.53 / 1.30 |
|  |  | Max | 0.47 / 0.44 / 0.32 | 0.36 / 0.32 / 0.22 | 1.83 / 1.52 / 1.42 |
|  | NWM | Linear | 0.94 / 0.90 / 0.86 | 0.91 / 0.85 / 0.78 | 0.42 / 0.31 / 0.20 |
|  |  | FFT | 0.84 / 0.78 / 0.65 | 0.76 / 0.67 / 0.52 | 1.04 / 0.68 / 0.51 |
|  |  | Last | 0.61 / 0.52 / 0.35 | 0.49 / 0.41 / 0.25 | 1.74 / 1.30 / 1.07 |
|  |  | Mean | 0.57 / 0.46 / 0.34 | 0.46 / 0.34 / 0.24 | 1.78 / 1.48 / 1.16 |
|  |  | Max | 0.59 / 0.46 / 0.41 | 0.47 / 0.34 / 0.31 | 1.55 / 1.47 / 1.11 |
| Tetrominoes | CNN | 2 | 0.26 / 0.26 / 0.00 | 0.16 / 0.16 / 0.00 | 3.50 / 1.55 / 1.54 |
|  |  | 4 | 0.40 / 0.39 / 0.00 | 0.27 / 0.25 / 0.00 | 3.55 / 1.30 / 1.28 |
|  |  | 8 | 0.83 / 0.82 / 0.00 | 0.72 / 0.70 / 0.00 | 3.55 / 0.38 / 0.33 |
|  |  | 16 | 1.00 / 0.00 / 0.00 | 1.00 / 0.00 / 0.00 | 3.61 / 3.50 / 0.01 |
|  |  | 32 | 1.00 / 0.00 / 0.00 | 1.00 / 0.00 / 0.00 | 3.60 / 3.40 / 0.01 |
|  | LSTM | Linear | 0.98 / 0.98 / 0.92 | 0.97 / 0.96 / 0.87 | 0.25 / 0.10 / 0.08 |
|  |  | FFT | 0.98 / 0.97 / 0.84 | 0.96 / 0.94 / 0.74 | 0.45 / 0.14 / 0.09 |
|  |  | Last | 0.92 / 0.84 / 0.00 | 0.86 / 0.73 / 0.00 | 3.55 / 0.43 / 0.29 |
|  |  | Mean | 0.96 / 0.68 / 0.00 | 0.93 / 0.53 / 0.00 | 3.61 / 0.72 / 0.20 |
|  |  | Max | 0.96 / 0.57 / 0.00 | 0.92 / 0.42 / 0.00 | 3.44 / 0.93 / 0.21 |
|  | NWM | Linear | 0.99 / 0.99 / 0.98 | 0.99 / 0.98 / 0.96 | 0.09 / 0.05 / 0.03 |
|  |  | FFT | 0.99 / 0.98 / 0.97 | 0.98 / 0.97 / 0.95 | 0.11 / 0.06 / 0.04 |
|  |  | Last | 0.98 / 0.97 / 0.81 | 0.97 / 0.95 / 0.70 | 0.53 / 0.15 / 0.10 |
|  |  | Mean | 0.99 / 0.94 / 0.78 | 0.98 / 0.89 / 0.65 | 0.50 / 0.23 / 0.06 |
|  |  | Max | 0.99 / 0.92 / 0.69 | 0.98 / 0.86 / 0.54 | 0.82 / 0.27 / 0.09 |

Table 6: Segmentation performance (only FG-Acc, FG-IoU, and FG-Loss). Values are max / median / min over 10 seeds.

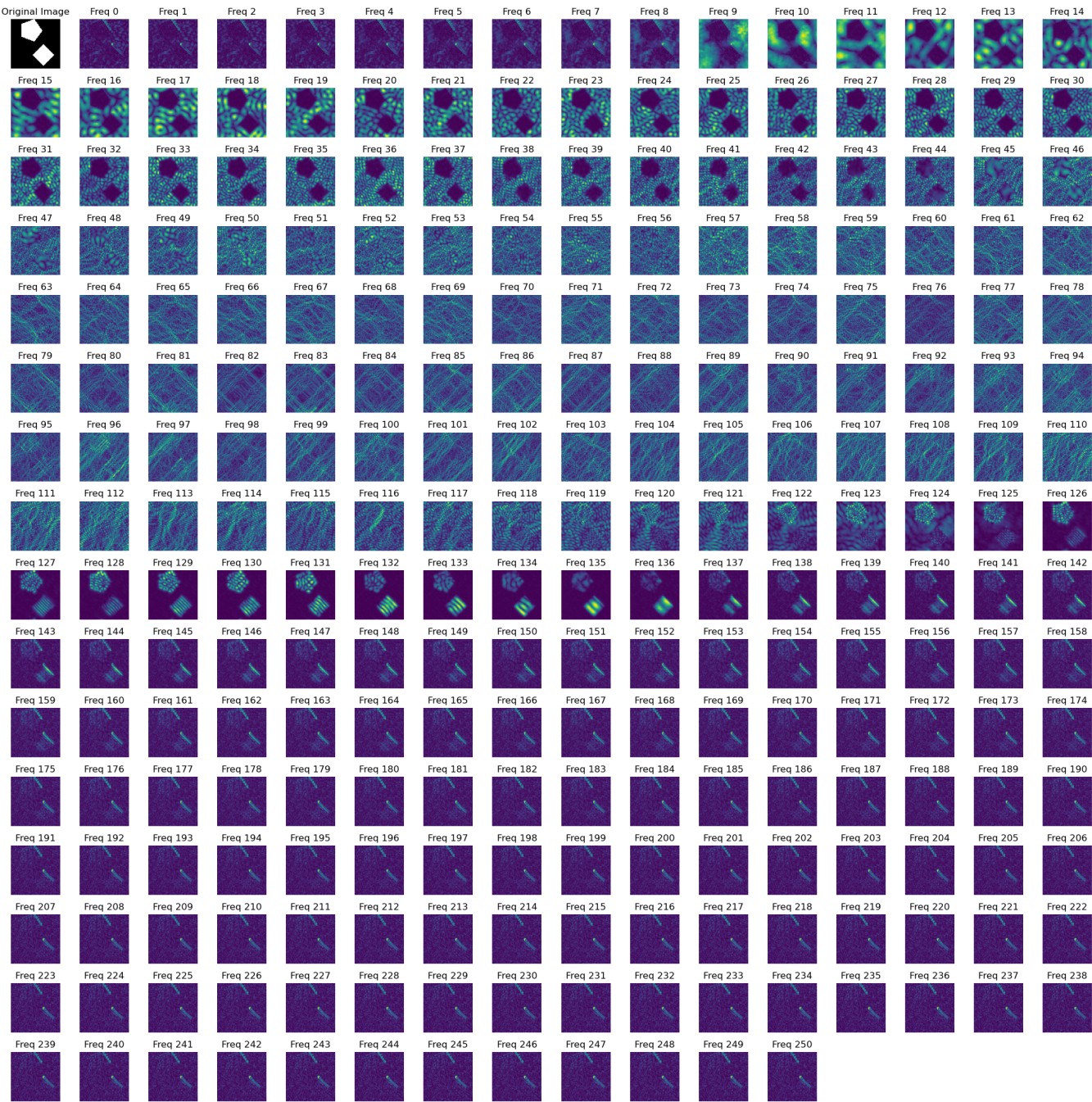

Figure 13: Visualization of all frequency bins for an example of the Polygons dataset. We see that the background and different shapes appear in separate frequency bins, allowing the model to easily segment the shapes semantically in frequency space.

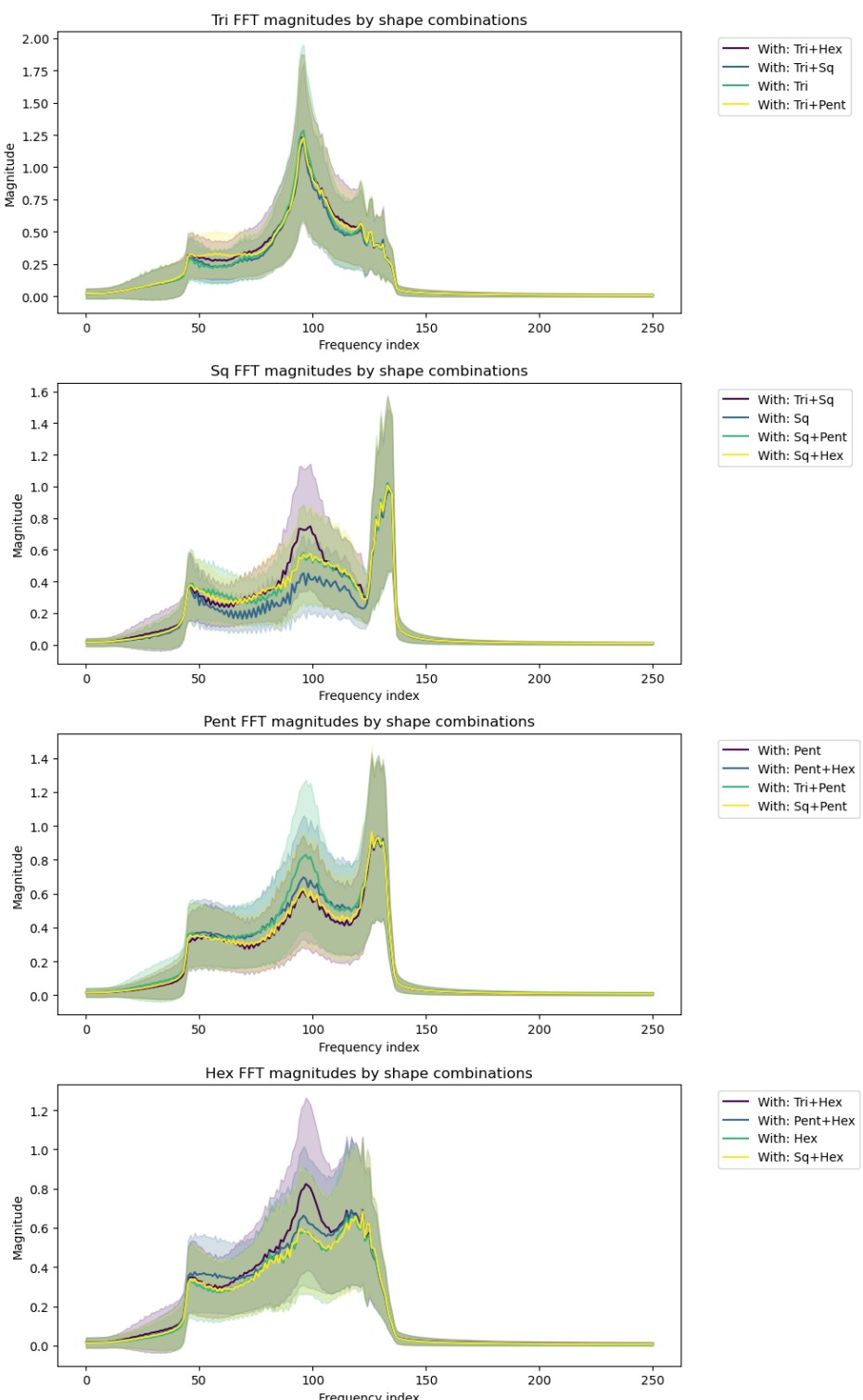

Figure 14: Visualization of the impact of different combinations of shapes in the same image on the frequency space representation of the other shape for the Polygons dataset. We see that while there is a minor impact on the frequency space representation of each shape when another shape appears nearby, the overall frequency spectrum is relatively invariant. This implies that each neuron indeed has global information about all shapes present in the image, but mainly represents the shape which it is currently 'located within'.

