# OpenReview forum: "Traveling Waves Integrate Spatial Information Through Time"
_ccneuro.org/CCN/2025/Proceedings — CCN 2025 Proceedings asProceedingsTalkPoster_

### Official Review · Reviewer_B5FP · 2025-03-23
**Promising approach to global information integration with recurrent dynamics; requires clearer presentation of methods**

**Soundness:** 3
**Clarity:** 3

**Comments:**

## Comments
I would like to congratulate the authors on a well-written paper with interesting and
exciting contributions to the discussion of global information integration from local
features. While I have criticisms of the current presentation of the work, I believe none are
fundamental and can be addressed, leading to substantial improvements in the
ratings I have given above.
This paper addresses an important problem: how to integrate information globally from
local feature detectors without requiring excessively deep CNN architectures. The
approach models each layer as a recurrent "CNN-RNN" with local, feedforward receptive
fields and recurrent dynamics that emulate the behavior of a vibrating membrane (e.g.,
drum skin). The paper gives a clear, new idea how recurrent dynamics can integrate
local information into global representations. The model is supported by multiple higher
-level measurements of visual cortex activity. The authors demonstrate this approach is
highly successful across various visual segmentation tasks, from simple geometric
shapes to segmenting multiple MNIST digits. While technically challenging to compute
due to the numerical simulation of oscillators, the approach demonstrates impressive
parameter efficiency.

Please find detailed comments on clarity and soundness of the work below.

### 1. Comments on clarity of the work

The authors present a relatively novel method (recurrent networks with wave dynamics)
and provide clear mathematical background through the "can you hear the shape of a drum
" analogy. However, the paper loses connection to this drum analogy when explaining the
actual networks being trained.

My understanding of the paper's logical flow (requesting clarification if incorrect) is:
1. Waves in a 2D medium with Dirichlet boundary conditions settle into standing wave
   solutions with corresponding modes and eigenvalues.
1. Previous mathematical work showed these solutions can identify boundary shapes.
1. The approach ports the discretized 2D wave equation to an RNN where neurons
   correspond to discretized spatial positions.
1. Dirichlet boundary conditions are implemented by fixing boundary neurons at 0.
1. The properties of the original wave equation solution are preserved in the RNN
   implementation.
1. The actual Neural Wave Machine model maintains these recurrent dynamics but handles
   input differently—becoming a network of oscillators where natural frequencies and
   dampening constants are determined by CNN functions of the input.
1. Here, the analogy to the original drum motivation seems lost, as the network
   segments shapes at different frequency components rather than performing mode/
   eigenvalue identification.
1. Best performance occurs when considering the entire time series, suggesting
   information exists in temporal evolution rather than steady-state solutions.

The paper allocates significant space to the intuitive drum analogy, creating the
impression that this mechanism is directly used for segmentation, which appears not to
be the case. I recommend using the "hear the shape of a drum" more clearly as motivation
, providing clearer explanation of the actual RNN implementation and its differences
from the motivating example, and offering an intuitive understanding for how input
influences the network via oscillator harmonics.

### 2. Comments on soundness of the work

The authors have conducted broad, well-chosen, and thorough control experiments. The
evidence as presented does seem to support all claims made, but the presentation is
sometimes unclear, which raises questions about validity. I believe these presentation
issues can be remedied, which would lead to very robust experimental support.

2.1) Experiments are presented clearly, though some minor details are missing in the main
text. The loss/validation metrics should be explained or at least mentioned in the text
. Results in the supplement hint at a metric that does not use the entire image but
only the objects in question - how is this implemented? It appears to be foreground
loss or foreground + background loss. Is there any danger of shifting from false
negatives to false positives with this approach?

2.2) In the supplementary results, has NWM become CoRNN? Is the NWM result without the
limitation to objects less impressive? This may not invalidate the findings, but it's
difficult to determine from the presented results. It is important that readers (
including myself) can easily contextualize the different results here! Similarly, the
complexity comparison is only provided for U-Net. While it need not be in all tables,
the relevant numbers should be presented more consistently, not just when the
comparison is favorable. I believe the paper's core argument may still be completely
valid, but the current presentation unnecessarily raises doubts about potential biases
in how the results are presented.

### 3. Minor comments and questions
3.1) If space permits, showing the new version of the MNIST task would help demonstrate the
capability of the paper.

3.2) Can you comment on the relevance of this high-level, abstracted description of network
dynamics as waves for neuroscience more generally? If an LSTM learns similar dynamics,
has it now become a wave LSTM? How about other RNN and SNN models? To what degree is
the very specific wave implementation needed for a model to show this behavior, or
could one see similar dynamics in RNNs that are clearly not modeling the physics of waves?

3.3) What other approaches exist to the fundamental question of integrating global
information from local features outside the wave-RNN world? To me, this is the core
contribution of the paper, and it doesn't have to live solely within the wave-RNN framework.

**Expertise:**

2

**Interest:**

3

---

> ### Author Rebuttal · Authors · 2025-04-15
>
> We thank the reviewer for their thoughtful feedback and highlighting where we can improve the paper’s clarity.
>
> We intend to emphasize a subtlety in the text clarifying the connection to the ‘drum’ analogy. In a coupled oscillator network (e.g. NWM), large differences in sub-population natural frequencies effectively decouple those groups, in some sense emulating a dirichlet boundary condition and causing reflection (as shown in Figure 3 where waves inside the polygon are isolated from those outside). This ‘soft’ boundary, created though varying natural frequencies, is one mechanism by which the model may instantiate a solution similar to ‘hearing the shape of the drum’. Moreover, each point’s oscillation frequency spectrum matches the eigenspectrum (Eq. 3). We have emphasized this is largely motivational in the text.
>
> In the main text, we focus on the foreground because the background is trivial for all models, inflating performance. A model that always predicts background gets 87% accuracy on MNIST and 86% on tetrominoes, yet 0% FG-ACC. We have added these ‘baseline’ values to the text and Supplementary Material (Figures 11, 12) to help contextualize the results. Given the standard segmentation metrics and high performance of models, we see results are consistent regardless of the accuracy metrics used. For completeness, Tables 3-6 of the Supp. details the full accuracy distribution.
>
> To our knowledge, the fundamental property of network architecture necessary to support wave-like dynamics is locally constrained recurrent connectivity. For related work, we point to (Keller et al., 2024b) for a sequence memory task and (Davis et al. 2021) for wave dynamics with a spiking neural network model.
>
> We agree that one of our core contributions is to bring attention to the question of how global information is integrated from local features; we also think any recurrent dynamics (even non-wave-like) could be mechanisms for integrating global information. Our paper can be seen as situated in literature exploring recurrence as an alternative to depth, but with a novel focus on traveling waves as the key operating mechanism and the finding of time-series readouts are crucial to performance.
>
> Some additions:
> - Listed evaluation metrics in the main text and supplement
> - Updated all tables to include # of parameters (omitted from our first draft due to space limitations, not because it was unfavorable)
> - Included figures of the new Multi-MNIST dataset in the supplement

---

> > ### Comment · Reviewer_B5FP · 2025-04-20
> >
> > Thank you for their extensive updates to your paper and considerations of my comments.
> > The updated version of the paper now includes very convincing support for their claims, and I think the overall paper has gained clarity through the updates as well.
> >
> > I must admit, I thought your explanation in your reply with regards to soft "drum" boundaries at object boundaries was even better than the text update, but thank you for clarifying the text anyways.
> >
> > I am satisfied and have updated my rating on clarity and, in particular, soundness of the work.

---

> > > ### Author Response · Authors · 2025-04-22
> > >
> > > Thank you for taking the time to go through our response and updated document. We truly appreciate the additions you have suggested and think the paper is much improved.

---

### Official Review · Reviewer_fKGe · 2025-03-28
**wave measurements**

**Soundness:** 2
**Clarity:** 2

**Comments:**

This work aims to show how traveling waves can help solve long-range visual segmentation problems. To do so it compares performance across neural wave machine models, convolutional LSTMs, feedforward CNNs, and U-nets on a variety of simple visual stimuli. It shows that by using a readout based on all timepoints, the NWM-based models can outperform models with more limited receptive fields. And can perform as well as parameter-heavy U-nets.

I don't feel entirely clear about the core contribution of this paper. I understand the claim that the NWM architecture is performing better but I am not clear on why. Specifically, the existence of traveling waves after training is not quantitatively analyzed as far as I can tell. The NWM is initialized to encourage traveling waves, but whether the weights remain similar to their initialization is not discussed. The authors point to Figure 5 to show the existence of traveling waves while solving the task, but personally I see stronger evidence of waves in the LSTM than the NWM. The authors also reference the fact that the LSTM can learn waves, so is the emergence of waves in the LSTM meant to be important for its function (the conclusion would suggest so, but it is not directly tested)? Figure 4 shows different frequencies for different objects but I don't know how to tie that directly to wave propagation. Is it possible to measure traveling wave strength directly and correlate it with performance? In any case, clarification on how exactly the wave dynamics help here would be useful. I think perhaps Figure 3 is meant to help with intuition, but by my eye the representation at the final time step as a result of the recurrent dynamics does not seem helpful for shape segmentation.

In terms of the LSTM and NWM comparison, how does the fact that the NWM get "inputs" that are already run through multi-layer CNNs get handled? The comparison to an LSTM getting pixel-level inputs wouldn't be fair, so does the LSTM also have several CNN layers on the front end before being passed into the recurrent layer?

**Expertise:**

2

**Interest:**

1

---

> ### Author Rebuttal · Authors · 2025-04-15
>
> We thank the reviewer for the valuable feedback, which helped us clarify our core contribution: **locally constrained neurons can transmit global information through traveling waves, providing a robust and biologically inspired mechanism for global integration**. This process was hypothesized in neuroscience but never mechanistically tested.
>
> In response to requests for more direct evidence, we added new quantitative analyses in Appendix Section “Extended Quantitative Analysis of Wave Dynamics” (Figures 9 and 10). Using the methods of Davis et al. [1], we track how wave properties (e.g., wavelengths) evolve during training and **observe a strong correlation between wavelength changes and model performance (R² = 0.961)**. This suggests that traveling waves are a meaningful conduit for global information.
>
> To clarify why recurrent models outperform CNNs with small receptive fields, Appendix Figure 7 illustrates how limited receptive fields may not fully capture target shapes, hindering global feature integration. By contrast, our locally recurrent models (NWM and LSTM) leverage wave dynamics to convey shape information across spatially distant regions—even with a small convolutional front end. We confirm that **both the NWM and LSTM share the same multi-layer CNN backbone**, isolating differences to their distinct recurrent architectures. While deeper CNNs achieve larger receptive fields, they often show higher variance and less stable training, possibly due to vanishing gradients or overfitting. In contrast, the oscillatory dynamics of wave-based models appear to provide greater stability.
>
> We thank the reviewer for highlighting confusion about Figure 3, which is central to our claims. As the reviewer pointed out, a single final timestep may not be useful for shape segmentation, but the collective dynamics (i.e., the full time series) encodes more comprehensive information. Table 1 shows that relying on just one timestep (“Arch. Last”) is less effective than harnessing the full time series (e.g., “Arch. FFT” or “Arch. Linear”). This is also how each neuron’s oscillation frequency connects to traveling waves: analyzing the entire timeseries is key to extracting shape-relevant signals.
>
> We appreciate the reviewer’s thoughtful comments and politely request they check the updated manuscript for details addressing their further concerns (e.g. Figure 8).
>
> [1] Davis et al. (2020). Spontaneous Traveling cortical waves gate perception in behaving primates. Nature.

---

> > ### Comment · Reviewer_fKGe · 2025-04-22
> >
> > I appreciate the authors' response; their additions help clarify things for me.

---

> > > ### Author Response · Authors · 2025-04-22
> > >
> > > We thank the reviewer for acknowledgment of our updates. We politely request, if they feel it is appropriate, to update their scores to reflect their updated opinion; and we thank them again for their time spent with our work.

---

### Official Review · Reviewer_dUAD · 2025-04-03
**Traveling waves improve performance on tasks requiring integration of information over space**

**Soundness:** 2
**Clarity:** 2

**Comments:**

This paper is concerned with the function of traveling waves in the brain and artificial neural networks. In particular, they test whether the inductive bias to produce traveling waves is good for performance in a variety of classification task that require integration of information across spatial domains larger than that accessible by individual units. They find that these waves are useful for computation. When they train a set of convolutional LSTMs, which are not biased to produce waves, they find that these emerge as a result of training.

This project has the potential to be of broad interest, but the paper is currently quite geared toward machine learning experts. The methods and analyses appear sound. However, the authors will need to provide more interpretation and intuition of the results in order to be clear to neuroscientists without ML background. It will also help to present graphical comparisons of the various models, as the tables and text demand a lot of the reader.

**Expertise:**

2

**Interest:**

3

---

> ### Author Rebuttal · Authors · 2025-04-15
>
> We thank the reviewer for their time spent with our paper and for their constructive feedback. We particularly appreciate the acknowledgement of the potential for broad interest in our work and the soundness of our analysis and methods.
>
>
> The reviewer's primary concern appears to be the readability of our work for neuroscientists who may not be familiar with the models or datasets. In response, we have added a number of 'finding specific' targeted graphs to the appendix, derived from the main tabular results, which illustrate the core findings in a much more reader friendly format. These plots can be found in the Appendix Section "Extended Visualizations of Table 1", and specifically the new Figures 7, 11, and 12.
>
>
> In detail, Figure 7 visualizes the size of the receptive field compared with the object feature sizes, demonstrating the need for the integration of information in order to solve the semantic segmentation task. Figure 11 provides a visual comparison of the NWM and LSTM performance across different readout metrics, highlighting the fact that FFT and Linear readouts over time significantly outperform the single element readouts such as Mean, Max and Last. Finally, Figure 12 directly compares the best performing recurrent models to CNN architectures of varying depths, demonstrating the high variance and low average performance of the deep CNN models compared with the recurrent (wave-based) alternatives.
>
>
> We thank the reviewer for this suggestion and welcome any further discussion following from these new figures.

---

> > ### Comment · Reviewer_dUAD · 2025-04-20
> >
> > The authors have provided a lot of helpful additional detail, particularly in Figures 7-11, which make the setup clearer and more clearly demonstrate the value of waves and their emergence during training.
> >
> > I still find the main text a bit oblique for the neuroscience field. I could imagine a framing, which more clearly states the following: deep nets perform well on visions tasks, these deep nets have units that look a bit like primate visual neurons, but real neurons have a lot of features that are inconsistent with deep nets, namely, recurrence and limited receptive field sizes. Those properties seem to make visual tasks more complex, but there may actually be advantages such as the long range integration of information via traveling waves. I dont want to impose this framing on the authors, and their work is already clear enough to be of interest, but hopefully it is helpful to hear a framing that makes things clearer for at least one neuroscientist.

---

> > > ### Author Response · Authors · 2025-04-22
> > >
> > > We thank the reviewer for their acknowledgment of our updated manuscript, and for appreciating the improvements. We politely request, if they feel it is appropriate, to update their scores to reflect their updated opinion; and we thank them again for their time spent with our work.
> > >
> > > We additionally truly appreciate the framing the reviewer has provided for the neuroscience community. We certainly agree with this line of reasoning, and think our work supports this as well. We will work this into the camera ready version of the paper and hope to make the work more readable to a broader audience.

---

### Meta-Review · Area_Chair_iPDd · 2025-05-06

**Ccn Recommendation:** Accept as Proceedings

**Metareview:**

This is a thoughtful and well-executed study that ties together insights from neuroscience, physics, and machine learning. The modeling approach is creative and well-justified, and the empirical results are compelling. All three reviewers were generally positive, raising useful  concerns that the authors addressed with clear updates.

Reviewer 1 suggested the framing was too ML-heavy. The authors responded with helpful visualizations (Appendix Figures 7, 11, 12) and committed to clearer exposition. Reviewer 2 asked for stronger evidence linking wave dynamics to performance; the authors added quantitative analyses (Appendix Figures 9-10) showing a strong correlation between wave properties and accuracy, and clarified architectural controls. Reviewer 3 praised the overall contribution but questioned how directly the “drum” analogy maps onto the results. The authors clarified its motivational role and provided additional explanation of the dynamics. The paper presents a rigorous and original approach to a core question in neural computation. The models are well-validated, and the insights into temporal readout and wave-based integration are valuable for both neuroscience and ML communities.

**Summary:**

This paper explores how traveling wave dynamics in recurrent neural networks can support global integration of spatial information, despite the local receptive field constraints that typically limit such architectures. Drawing inspiration from the “drum” analogy, the authors propose wave-biased recurrent models (Neural Wave Machine, or NWM) and evaluate them on a set of visual segmentation tasks. They show that models leveraging full time-series readouts can match or outperform deeper feedforward networks, including U-Nets, while using fewer parameters and exhibiting greater training stability. The findings suggest that traveling waves may provide a computationally efficient and biologically plausible mechanism for long-range visual integration.

**Expertise:**

2